# How the banking system is creating a two-way inflation in an economy

Ahmed Mehedi Nizam *

The Central Bank of Bangladesh, Motijheel, Dhaka, Bangladesh

* ahmed.mehedi.nizam@gmail.com

## Abstract

Here we argue that due to the difference between real GDP growth rate and nominal deposit rate, a demand pull inflation is induced into the economy. On the other hand, due to the difference between real GDP growth rate and nominal lending rate, a cost push inflation is created. We compare the performance of our model to the Fisherian one by using Toda and Yamamoto approach of testing Granger Causality in the context of non-stationary data. We then use ARDL Bounds Testing approach to cross-check the results obtained from T-Y approach.

**Data Availability Statement:** All relevant data are within the paper and its Supporting Information files.

**Funding:** The author received no specific funding for this work.

## 1 Introduction

We propose a new model that describes the role of the banking system in creating a two-way inflation in an economy. According to the proposed model, when the nominal deposit interest rate of the bank is set to a value which is higher than the underlying real GDP growth rate then the money in the depositors' account grows faster than the goods in the real sector. So, it will lead to *too much money chasing too few goods* type of scenario which eventually shifts the aggregate demand curve upward. Upward shift of the aggregate demand curve results into a demand pull inflation and an inflationary gap in output. On the contrary, when the nominal deposit interest rate is lower than the underlying real GDP growth rate then the money in the depositors' accounts grows slower than the goods in the real sector which increases the purchasing power of the money and thereby decreases the general price level by shifting the aggregate demand curve downward and we have a recessionary gap in output as a by-product.

On the other hand, when the borrowers (investors) are charged at a rate higher than the real GDP growth rate, they (borrowers/investors) have to pay more money than they actually earn by investing the borrowed fund into the real sector. As interest expense is usually considered to be a cost of production (see for example, [1], [2], [3], [4] among others), a rise in interest expense on per unit of produced goods results into an upward shift of the aggregate supply curve. As the supply curve shifts upward, equilibrium is achieved at a higher general price level resulting into a cost-push inflation and a recessionary gap in output. The opposite holds true also. When the economy grows at a rate higher than the nominal lending rate charged by the bank then the borrowed fund injected into the economy will earn more than it costs. Thus, interest expense of the leveraged business concerns are compensated by the rapid growth of

**Competing interests:** The author has declared that no competing interests exist.

the economy and interest expense on per unit of produced goods decreases resulting into a downward shift of the aggregate supply curve. A downward shift of the aggregate supply curve is then translated into a decrease in general price level and an inflationary gap in output. Both the recessionary and inflationary gap in output eventually shrinks and the output converges to its original long run full employment level with a different price tag.

Apart from nominal deposit and lending rate, we also consider the total volume of deposit and credit in the banking system in establishing the relationship between interest rate and inflation. Because, if the amount of deposit and credit in the banking channel is not substantial as compared to the overall size of the economy then the causality running from nominal interest rate to inflation becomes weak. Here, we try to quantify the combined impact of the aforementioned variables on the inflation and provide two metrics which, according to our point of view, can be linked to inflation. The rest of this paper is organized as follows: Section 2 describes the rational behind adopting a new model that relates nominal interest rate and inflation, Section 3 & 4 show how nominal deposit & lending rate can induce a demand pull & cost push inflation respectively. Section 5 determines the combined effect of nominal deposit and lending rate on inflation in the short run. Section: 6 describes the long run self adjustment mechanism. Section 7 explains the methodology used to statistically verify our claim. Section 8 presents the data obtained in statistical analysis. Section 9 compares the result of our model to that of the Fisherian one and finally, Section 10 makes some concluding remarks.

## 2 Rational behind adopting a new model

The only well known and most studied relationship between interest rate and inflation is the so-called Fisher Hypothesis [5] which says that the nominal interest rate rises point-for-point basis with the expected inflation assuming the real interest rate to be constant. Since its inception in 1930, a number of empirical studies have been carried out to judge its effectiveness in describing the relationship between interest rate and inflation and the results of these vast amount of empirical analysis are mixed: Some studies find the evidence of Fisherian link while the others reject it. Atkins (1989) [6] has shown that the post-tax nominal interest rates and inflation in Australia and USA for the period 1953-1981 are cointegrated in the sense of Engle and Granger and these variables have a joint error correction representation. Findings of Atkins (1989) [6] suggest existence of long run Fisher Effect in the aforementioned economies for the designated period. However, using the same Engle-Granger approach of cointegration, Macdonald and Murphy (1989) [7] have found no evidence of Fisher Effect in the data of USA, UK, Canada and Belgium for the period 1955-1986. Macdonald and Murphy (1989) [7] then divide the data depending upon the exchange rate regime and in the modified experimental set-up they have found evidence of Fisherian link only for USA and Canada. Moreover, Dutt and Ghosh (1995) [8] investigate the validity of the Fisher Effect under both fixed and floating exchange rate regime. Johansen test of cointegration methodology is applied to test the weak form of Fisher Effect while Phillips-Hansen fully modified ordinary least square (FM-OLS) technique is applied to test the strong form of Fisher Effect. However, in both cases and in both fixed and floating exchange rate regimes, the Fisher Effect is soundly rejected. But, Crowder (1997) [9] has found significant evidence of the existence of Fisher Effect in Canadian data of inflation and nominal interest rate although the Fisherian relationship is not found to be stable in the period examined. Crowder and Hoffman (1996) [10] also find evidence of tax adjusted Fisher Effect on the US and Canadian data using Johansen Test of co-integration. Meanwhile, Fahmy and Kandil (2003) [11] observe that inflation and nominal interest rate exhibit common stochastic trend in the long run. But, in the short run, no common trend is observed which implies there is no such Fisher effect in the short run.

All the above approaches uses the concept of cointegration in one form or another and cointegration requires each of the variables under consideration to be of I(1): Variables must be stationary at first difference, but non-stationary at level. So, we need some form of robust test for the presence of unit root in time series before we go for checking cointegration and none of the standardized tests of checking stationarity of time series is that much robust. Different tests of stationarity or even the same test with different parameter setting may give different results regarding the order of integration of the time series under consideration [12]. So, the success of all the above literature critically depends on determining the correct order of integration of the time series. To overcome this difficulty, Frank J. Atkins, Patrick J. Coe (2002) [12] have applied the ARDL Bounds testing approach developed by Pesaran, Shin and Smith (2001) [13] to study the existence of long run cointegrating relationship between nominal interest rate and inflation. ARDL Bounds Testing approach can be comfortably applied to the data which can be any mixture of I(0) and I(1) processes. Their results do not support tax adjusted Fisher Effect for Canada during the period 1953-1999 and for the US data in the same period, their conclusion regarding the existence of the so-called Fisher Effect is somewhat in the grey region. However, Koustas and Serletis (1999) [14] apply King and Watson (1997) [15] methodology to test Fisher Effect in the post-war quarterly data of nine industrialized country (Belgium, Canada, Denmark, France, Germany, Greece, Ireland, Japan, the Netherlands, the United Kingdom and the United States) and they eventually find no evidence in favor of Fisher Effect.

Another strand of literature analyzes the role of the estimators used in empirical validation of Fisher hypothesis. Caporale and Pittis (2004) [16] argue that the validation or rejection of Fisher effect in empirical literature critically depends upon the estimators used in the analysis. They show that the estimators most commonly used in the literature, namely OLS, Dynamic OLS (DOLS) have worst performance in small sample and usually reject the Fisher hypothesis. However, using US data, they have shown that if one employs estimators with smallest downward bias and minimum shift in the distribution of associated t-statistics it is highly likely for the Fisher hypothesis to be empirically accepted. Westerlund (2007) [17] has shown that rejection of Fisher hypothesis in empirical literature is partly due to the low power of univariate tests and proposes two panel cointegration tests which can be applied under very general condition. Westerlund (2007) [17] applies the proposed panel cointegration tests upon a panel of quarterly data of 20 OECD countries and provides evidences in support of Fisher effect.

A new cluster of research tries to investigate the inter-relation between interest rate and inflation by considering non-linearities in the equilibrium relationship (see Panopoulou and Pantelidis (2016) [18]). Bierens (2000) [19] has observed that interest rate and inflation share common non-linear trends. Lanne (2006) [20] introduces a nonlinear bivariate mixture autoregressive model that seems to fit quarterly US data during 1953 -2004. Koustas and Lamarche (2010) [21] have shown that ex-post real interest rates follow a nonlinear model characterized by mean reversion and provide statistical evidence for the Fisher effect. Christopoulos and Leon-Ledesma (2007) [22] argue that the empirical failure of the Fisher effect is due to the existence of non-linearities in the long run relationship between interest rates and inflation and present evidence that the Fisher relation presents important non-linearities for US data during 1960-2004.

All the exhaustive literature mentioned above hinges around the empirical verification of the Fisher Effect in different set up and varying time frame or try to gauge the goodness of the estimators used in the analysis with no attempt to augment the Fisherian model with some core elements it has been missing. From our point of view, Fisher Effect, albeit elegant, is too simple to be true. First of all, it overlooks the impact of contemporary real GDP growth rate while establishing the long run relationship between interest rate and inflation. As we have

already mentioned in the introductory section of this article, the difference between real GDP growth rate and nominal deposit rate can give birth to demand pull inflation (deflation) in the economy. When the nominal deposit rate is higher than the real GDP growth rate then the money in the depositors' accounts grow faster than the goods in the real sector and it leads to a situation where *too much money is chasing too few goods* and vice versa. On the other hand, when the nominal lending rate is set to a value which is higher than the contemporary real GDP growth rate then the borrowers (investors) have to pay more money than they actually earn by investing it (the borrowed fund) into the real sector which results into an upward shift in aggregate supply curve. This eventually creates a cost push inflation in the economy. Secondly, the Fisher Effect does not discriminate between two different types of interest rate namely, deposit interest rate and lending interest rate, which may effect inflation in different ways. As we have mentioned previously, the deposit interest rate is tied to demand pull inflation while the lending interest rate is tied to the cost push one: One intends to shift the aggregate demand curve upward while the other raises the general price level by pushing up the aggregate supply curve. Fisher Effect, being overly simplified, does not make any mention to these two very different forms of inflation existing in the economy who are inherently different from their point of origin. Next, Fisher Effect fails to account for the volume of deposit and credit which, from our point of view, can not be ignored. When the size of the deposit (credit) is insignificant as compared to the total GDP of the economy, the effect of interest rate on inflation will be negligible. This is because, when the amount of deposit (credit) is insignificant then it will effect only a handful of people in the economy and thereby its effect on the general price level would be insignificant. On the other hand, when the amount of deposit (credit) is comparable to the GDP of the economy then the effect of interest rate (both deposit and lending interest rate) on inflation will be very much pronounced. One last point about the Fisher Effect, although it algebraically relates the interest rate and inflation, it mostly ignores the overall macro-economic mechanism that links them together. The points aforementioned encourages us to provide a new model that more clearly captures the dynamic relationship between interest rate and inflation and shed some light on the macro-economic mechanism that holds them together.

## 3 Relationship between inflation and nominal deposit rate

Let, $d$ be the nominal deposit rate, $g$ be the real GDP growth rate and $D$ be the total amount of deposit in the banking system.

Then the total amount of nominal interest income received annually by the depositors is given the following construct:

$$d \times D$$

If the nominal deposit rate $d$ becomes equal to real GDP growth rate $g$ then money in the depositors' accounts (i.e., the cumulative savings of the depositors) grows at the same pace as the goods grow in the real sector. Equivalently, we can say if depositors' cumulative savings grow at the same pace as the goods grow in the real sector then output to cumulative savings ratio remains the same over the years as both the growth factors cancel out each other:

$$\frac{G}{D} = \frac{(1+g) \times G}{(1+g) \times D}$$

where $g$ is the growth factor and $G$ is the output in nominal terms. Depositors in this case tend to spend the same amount of money on each unit of produced goods as both goods and depositors' cumulative savings grow equally over the time. Nominal interest income thus received

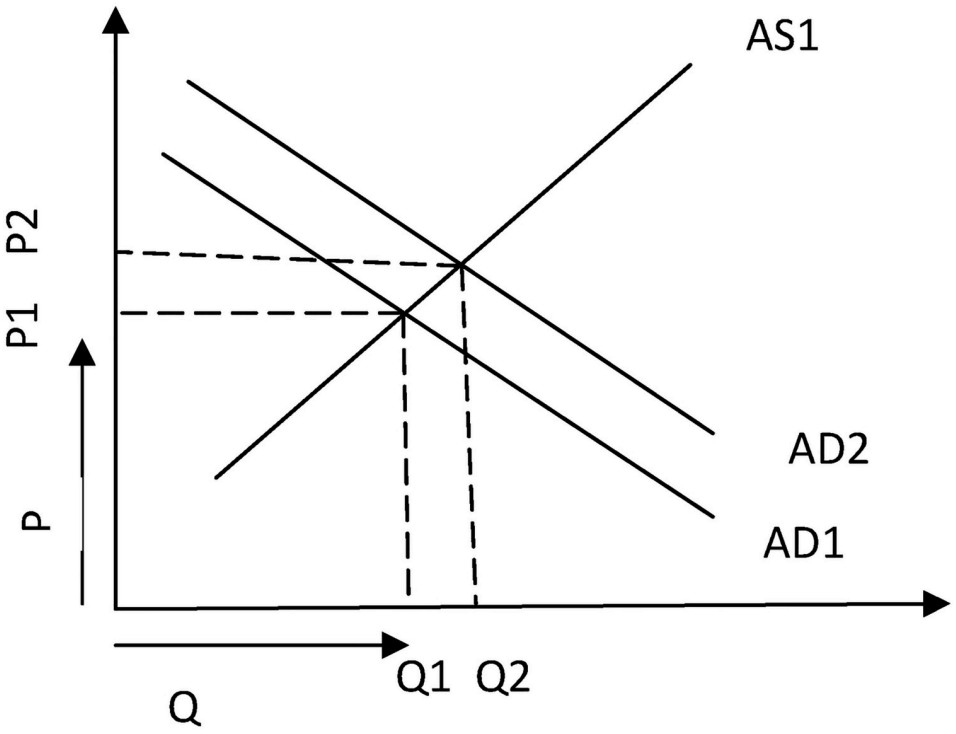

**Fig 1. Impact of interest income on aggregate demand.**

annually by the depositors in this case is given by:

$$g \times D$$

Any nominal interest income above and beyond $g \times D$ will increase the depositors' ability to spend more money on goods and services which makes the aggregate demand curve move upward in the short run. These dynamics are graphically represented in Fig 1.

This increase in depositors' ability to spend more money on goods and services can be quantified by the following construct:

$$d \times D - g \times D$$
$$= (d - g) \times D$$

The above quantity represents a portion of nominal interest income received by the depositors which are not supported by an equivalent increase in goods and services in the real sector. A portion of this *extra* nominal interest income will be spent while the other portion will be saved. If the average propensity to consume is given by *APC* then the portion of *extra* nominal interest income spent by the depositors on goods and services is given by:

$$APC \times (d - g) \times D$$

If the nominal GDP of the economy is given by $G$ then the amount of *extra* nominal interest income spent by the depositors on each unit of produced goods is given by:

$$\frac{APC \times (d - g) \times D}{G} \tag{1}$$

The last quantity will be our metric to quantify the extent of demand pull inflation caused by the banking channel. We name this quantity as *extra* amount of nominal interest income the depositors pay on each unit of consumed goods and services. It is so named as it represents only a 'monetary' increase which is not backed by an equivalent growth in the real sector.

## 4 Relationship between inflation and nominal lending rate

Let, $l$ be the nominal lending rate, $g$ be the real GDP growth rate and $L$ be the total amount of credit in the banking system.

Then the nominal interest expense incurred by the borrower is given by:

$$l \times L$$

On the other hand, when the economy is growing at a rate $g$, we can assume that the producers of goods and services as a whole also get a $g$ percentage point growth in their production, revenue and profit. If loans in the borrowers' account and output in real terms grow at the same pace (i.e., if $g = l$) then the accruals in loans can be served effectively by the enhancement in profit. If however $g < l$ then the borrowers have to manage extra money for interest servicing which can not be obtained from the growth in profit and as this is an economy-wide phenomenon not just for one single producer, the aggregate supply curve moves upward consequently. On the other hand, when $g > l$, the opposite holds true: The aggregate supply curve moves downward and a lower equilibrium price level is set in the short run. These phenomena can be pictorially depicted in Fig 2.

So, up to $g \times L$ amount of interest expense can be effectively served by the borrowers from their growth in production, revenue and profit. Volume of interest expense above and beyond

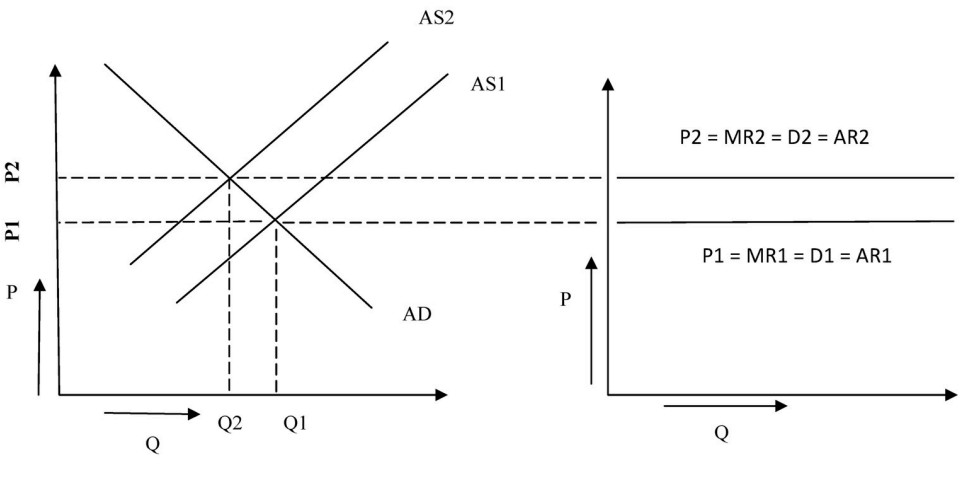

**Fig 2. Impact of interest expense on macroeconomy and individual firm.**

$g \times L$ is given by the following construct:

$$(l - g) \times L$$

As the above volume of interest expense must be served from selling the total goods and services produced, the amount of extra interest expense attributed to per unit of goods and services produced will be given by the following construct:

$$\frac{(l - g) \times L}{G} \tag{2}$$

The last quantity will be our metric to quantify the extent of cost push inflation caused by the banking channel. We name this quantity as *extra* amount of nominal interest expense incurred by the borrowers on each unit of produced goods and services. It is so named as it represents only a 'monetary' increase which is not backed by the corresponding growth in the real sector.

## 5 Combined effect of nominal deposit rate and nominal lending rate on inflation in the short run

Prevoiously we calculate the impact of nominal deposit rate and nominal lending rate on inflation individually. Here we will calculate the combined impact of these two rates on inflation in the short run. To do so, we first divide the depositors into 2 classes: One class of depositors have only deposit but no loan with the bank while other type of depositors have both deposit and loan with the bank. Let us assume that $\alpha, 0 \leq \alpha \leq 1$ be the portion of deposit whose owners do not have loan accounts with the bank. So, $(1 - \alpha)$ will be portion of deposit whose owners have both loan and deposit account with the bank. We also assume that $\beta, 0 \leq \beta \leq 1$ is portion of credit of those borrowers who do not have deposits with the bank. So, $(1 - \beta)$ will be the portion of credit of those borrowers who have both deposits and credits with the bank.

So the *extra* amount of nominal interest income spent on per unit of goods by the depositors who do not have credit with the bank is given by the following construct.

$$\frac{\alpha \times APC \times (d - g) \times D}{G} \tag{3}$$

On the other hand, the *extra* amount of nominal interest expense paid by the borrowers on per unit of goods produced who do not have deposits with the bank, will be given by the following expression.

$$\frac{\beta \times (l - g) \times L}{G} \tag{4}$$

Remaining $(1 - \alpha)$ portion of deposits is owned by the customers who have borrowed $(1 - \beta)$ portion of the total loan. Whether this segment of customers get or pay more money over and above the real GDP growth, will depend upon the sign of the following quantity.

$$\frac{(1 - \alpha) \times (d - g) \times D}{G} - \frac{(1 - \beta) \times (l - g) \times L}{G} \tag{5}$$

If the sign of the above quantity is positive then the segment of customers who have both loan and deposit with the bank will receive more money than they pay for their loan and the difference between amount received & amount paid, will cause aggregate demand curve move upward and therefore, a demand pull inflation will be created. So combining the contribution of these two segements of customers (who have only deposit and who have both deposit &

loan), we find overall *extra* amount of nominal interest income spent on per unit of goods produced (*EM*) which will be given by the following equation:

$$EM = APC \times \left( \frac{\alpha \times (d-g) \times D}{G} + \left( \frac{(1-\alpha) \times (d-g) \times D}{G} - \frac{(1-\beta) \times (l-g) \times L}{G} \right) \right)$$

$$EM = APC \times \left( \frac{(d-g) \times D}{G} - \frac{(1-\beta) \times (l-g) \times L}{G} \right) \tag{6}$$

In this case, the total amount of *extra* nominal interest expense incurred by the customers who borrow to produce, will be given by the construct given in Eq 4.

However, if the sign of the quantity given in Eq 5 is negative then the segment of customers who have both deposit and loan accounts, will pay more money than they recieve on top of the real GDP growth. So, then the overall amount of *extra* nominal interest expense incurred by the two segments of customers (one who have only loan and the one who have both loan & deposit with the bank) to produce per unit of goods will be given by:

$$EC = \frac{\beta \times (l-g) \times L}{G} + \left( \frac{(1-\beta) \times (l-g) \times L}{G} - \frac{(1-\alpha) \times (d-g) \times D}{G} \right)$$

$$EC = \frac{(l-g) \times L}{G} - \frac{(1-\alpha) \times (d-g) \times D}{G} \tag{7}$$

In this case, the *extra* amount of nominal interest income spent by the depositors on each unit of produced goods and services will be given by construct given in Eq 3.

The *extra* amount of nominal interest income the depositors spend on per unit of produced goods (EM) will shift the demand curve upward while the *extra* amount of nominal interest expense (EC) incurred by the borrowers will shift the supply curve upward. The whole dynamics are graphically represented in Fig 3.

Now, let us assume a parallel shift of demand and supply curve by an amount $d_1$ and $d_2$ respectively. Let us also assume that, initially, the demand and supply curve are given by the following two equations:

$$P = m_d \times Q + c_1$$
$$P = m_s \times Q + c_2$$

Let the shifted set of equations are given by:

$$P = m_d \times Q + c_3$$
$$P = m_s \times Q + c_4$$

In the above equations, $m_d$ and $m_s$ are the slope of demand and supply curve. As we assume parallel shifts in demand and supply curve, $m_d$ and $m_s$ remain unchanged in the shifted equations. Then using simple geometric analysis, it can be shown that the change in price ($\Delta P$) in response to the shifts in demand and supply curve is given by the following:

$$\Delta P = \frac{m_d}{m_d - m_s} \times d_2 \times sec(\theta_2) + \frac{m_s}{m_d - m_s} \times d_1 \times sec(\theta_1) \tag{8}$$

where $\theta_1$ and $\theta_2$ are the angle of inclination of demand and supply curve respectively. As we only assume parallel shifts, $m_d$, $m_s$, $\theta_1$ and $\theta_2$ remain unchanged. So, the above equation turns

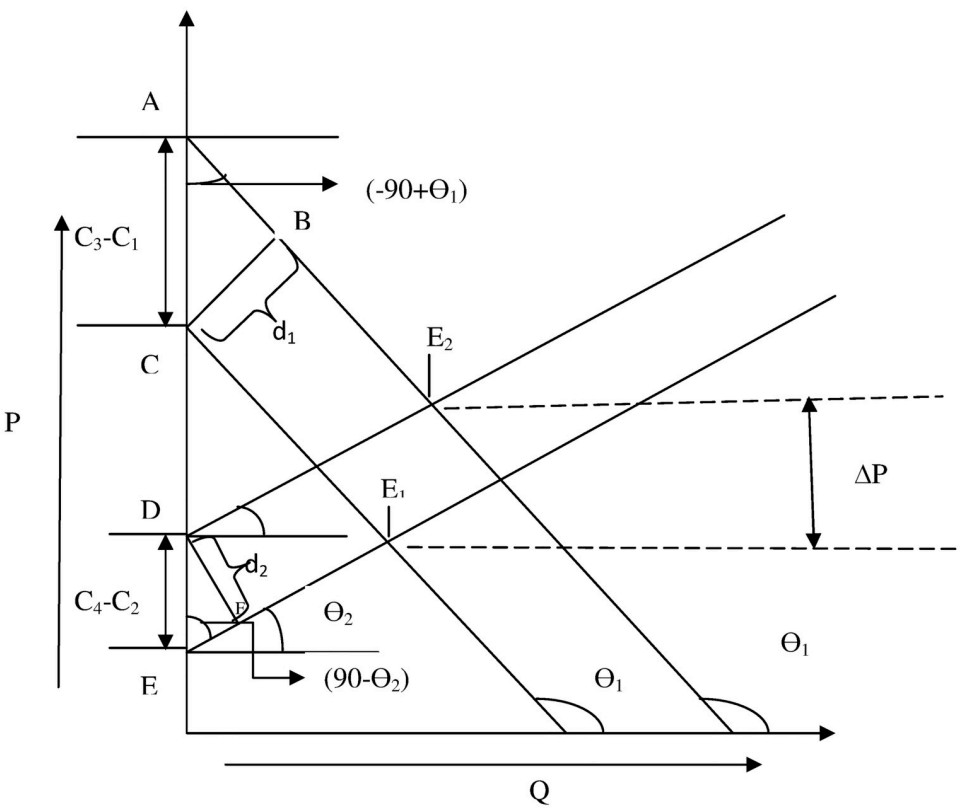

**Fig 3. Impact of *EM* and *EC* on demand and supply.**

out to be:

$$\Delta P = K_1 \times d_1 + K_2 \times d_2$$

Where $K_1$ and $K_2$ are constants. As we mentioned previously, $d_1$ and $d_2$ are the parallel shifts of demand and supply curve, they will depend upon *EM* and *EC*. Higher the value of *EM* and *EC*, higher will be the value of $d_1$ and $d_2$ respectively. So, we can safely assume that $d_1$ & $d_2$ are proportional to *EM* & *EC* respectively. Considering this, we can rewrite the above equation as follows:

$$\Delta P = K_3 \times EM + K_4 \times EC \tag{9}$$

Where $K_3$ and $K_4$ are constants. Now, if the sign of the quantity given in Eq 5 is positive then we can substitute the value of *EM* and *EC* from Eqs 6 and 4 into Eq 9. Then we get the following equation that relates change in price ($\Delta P$) to *EM* and *EC*:

$$\Delta P = \frac{K_3 \times APC \times (d-g) \times D}{G} + \frac{(K_4 \times \beta - K_3 \times (1-\beta) \times APC) \times (l-g) \times L}{G} \tag{10}$$

So,

$$\frac{\Delta P}{P} = \frac{K_3 \times APC \times (d-g) \times D}{P \times G} + \frac{(K_4 \times \beta - K_3 \times (1-\beta) \times APC) \times (l-g) \times L}{P \times G} \tag{11}$$

But, if the sign of the quantity given in Eq 5 is negative then we substitute the value of *EM* and *EC* from Eqs 3 and 7 into Eq 9. And, we get the following after simplification:

$$\Delta P = \frac{(K_3 \times \alpha \times APC - K_4 \times (1 - \alpha)) \times (d - g) \times D}{G} + \frac{K_4 \times (l - g) \times L}{G} \tag{12}$$

So,

$$\frac{\Delta P}{P} = \frac{(K_3 \times \alpha \times APC - K_4 \times (1 - \alpha)) \times (d - g) \times D}{P \times G} + \frac{K_4 \times (l - g) \times L}{P \times G} \tag{13}$$

## 6 Long run self adjustments

As interest rates and other variables involving *EM* and *EC* fluctuate over the course of time, changes in AD and AS in response to changes in *EM* and *EC* are rather transitory in nature. So, an inflationary/recessionary gap in output may be created in short run due to sticky prices. But, in the long run, prices adjust and the economy goes back to its full employment level with a change in general price level. Here, two different cases may occur: We either have an inflationary gap when AD and SRAS curve intersects on the right hand side of LRAS curve or we might have a recessionary gap when AD and SRAS curve intersects on the left hand side of LRAS curve. Two cases along with their eventual self adjustments are pictorially depicted in Fig 4.

From the left hand side of Fig 4, we can see that initially the economy was in its long run equilibrium at point $(Q_1, P_1)$. But, due to the demand and supply shock brough about by *EM* and *EC*, a short run equilibrium is achieved at $(Q_2, P_2)$ on the right hand side of LRAS curve which corresponds to a positive output gap. It means that the economy is overheated, unemployment is lower than its natural rate and the price level is higher than its original equilibrium one. So, wide-spread inflation makes the prices of the factors of production adjust above their initial values. As the costs of the factors of production rises, so does the cost of production itself. Hence, aggregate supply curve shifts upward and continues to do so until and unless the economy is brought back to its original full employment level but with a higher general price tag than before. The new long run equilibrium is achieved at point $(Q_1, P_3)$. As evident from Fig 4, $P_3 > P_1$. It means the long run equilibrium is achieved at a higher general price level.

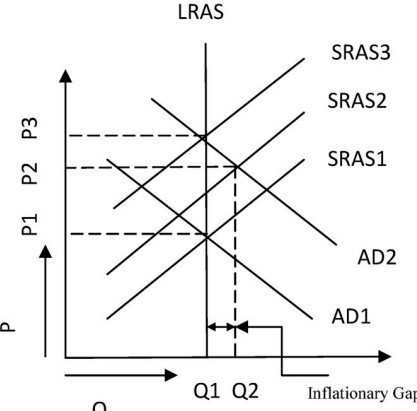

Closing down an inflationary output gap

Closing down a recessionary output gap

**Fig 4. Mechanism of long run self adjustments.**

From the right hand side of Fig 4, we can see the formation of a recessionary gap in output due to changes in AD and AS. Changes in AD and AS are brought about by *EM* and *EC* respectively. Here, the initial long run equilibrium correspoonds to the point $(Q_1, P_1)$ and the short run equilibrium established after shock corresponds to the point $(Q_2, P_2)$ on the graph. As there is a recessionary gap, the economy is now under performing and unemployment is higher than its natural rate. As there is less scope for work, cost of labour, i.e., wages along with the cost of other factors of production adjust downward. Lower adjustment of wages, rent, cost of capital makes the SRAS curve moves downward until and unless it intersects the LRAS curve again. Hence, the economy is brought back again to its initial full employment level with a different price tag than before and the new equilibrium corresponds to the point $(Q_1, P_3)$ on the graph.

## 7 Methodology

We can see from Eqs 11 and 13, in all cases (irrespective of the sign of the quantity given in Eq 5 is positive or negative), inflation is some linear combination of the constructs given in Eqs 1 and 2. So, we build a model where inflation is the dependent variable and the quantities given in Eqs 1 & 2 namely, $\frac{APC \times (d-g) \times D}{GDP}$ and $\frac{(l-g) \times L}{GDP}$ are the two independent variables.

On the other hand, to model Fisher effect, we invoke rational expectation and assume the expected inflation at any point of time, is given by the actual inflation one period ahead of the present time. Assuming this, the Fisher equation turns out to be:

$$i_t = A_0 + A_1 \times \pi_{t+1} + \epsilon_t$$

where $i_t$ is the nominal interest rate at time $t$, $\pi_{t+1}$ is the expected inflation at time $t$ which is the actual inflation at time $(t + 1)$ and $A_0$, $A_1$ are constants, $\epsilon_t$ is the error term. We use nominal lending interest rate to model nominal interest rate and annual GDP deflator to model inflation.

If the Fisherian equation succeeds as an algebraic equality then it must confer the following two things among others:

- Inflation and (time lagged) interest rate are cointegrated.

- There must have been a bidirectional causality running amongst the aforesaid variables.

The above two statements provide us a solid ground upon which we can empirically compare the performance of our model to the Fisherian one. To do so, the following steps are followed:

- We begin our analysis by testing for unit roots in the underlying time series. Five different time series namely, inflation, nominal deposit rate, nominal lending rate, $\frac{APC \times (d-g) \times D}{GDP}$ and $\frac{(l-g) \times L}{GDP}$ of five OECD countries (Australia, Japan, Korea, Switzerland and UK) are tested for the presence of unit roots using Augmented Dickey Fuller (ADF) Unit Root Test. The countries are arbitrarily chosen depending upon the availability of data. As we know, the ADF test comes up with different variants: 1) having intercept only 2) having trend and intercept and 3) no trend, no intercept in the equation, all these variants are tested.

- One of the most popular approaches of testing Granger non-causality in the context of non-stationary time series is the T-Y approach proposed by Toda and Yamamoto [23]. Here, we recall that our proposed model confers a causal relationship running from two of our metrics namely $\frac{APC \times (d-g) \times D}{GDP}$ and $\frac{(l-g) \times L}{GDP}$ to inflation. On the other hand, Fisher equation being an algebraic identity posits a bidirectional causality running between expected inflation and current

lending rate. We employ T-Y approach in testing all the aforementioned causal relationships which might exist in the empirical data. Steps to be followed for T-Y approach are depicted in the apppendix.

- After causality is conferred by the T-Y procedure, we can cross check the result by performing cointegration test amongst the same set of variables. If there is cointegration amongst the variables, then there must exist causality in either direction or both. In order to cross check the result obtained at the previous step, we will check for cointegration using ARDL Bounds Testing approach proposed by Pesaran, Shin and Smith (2001) [13]. This is indeed a special kind of cointegration testing that is intended to handle both $I(0)$ and $I(1)$ variables simultaneously. Unlike other popular approaches of testing cointegration like the Johansen Test of Cointegration, ARDL Bounds Testing approach can be applied to any combination of $I(0)$ and $I(1)$ variables which made it a more generic choice.

## 8 Data

We collect annual data of nominal lending rate, nominal deposit rate, inflation (GDP deflator), money supply (M2) as percentage of GDP, domestic credit provided by the financial sector as a percentage of GDP and gross savings as a percentage of GDP from World Bank data warehouse which is publicly available through the URL: data.worldbank.org/indicator. To ensure consistency among datasets, we only use data from that single source. We approximate the total deposit of the banking sector by the Broad Money (M2) on the ground that Broad Money (M2) is positively correlated to the banks' total demand and time liabilities. Average Propensity to Consume (APC) is measured by (1-gross savings as a perentage of GDP). The sampling period is from 1960 to 2014 although some series are truncated (listwise deletion) between this range depending upon the availability of the data. Data of some 5 (five) OECD countries are collected and analyzed. Countries are chosen by the availability of the data.

### 8.1 ADF Unit Root Test and the value of *m* for T-Y procedure

The results obtained by performing ADF Unit Root Test are presented in Tables 1, 2, 3, 4 and 5. From these tables, the value of *m* (the maximum order of integration of any group for any country) can be determined. It is revealed from these tables that the value of *m* for our proposed model is: 1 (one) for Australia & Switzerland, 0 (zero) for Japan & Korea and 2 (two) for UK while for Fisherian Model, the value of *m* is: 1 for Australia, Japan, Korea & Switzerland and 2 (two) for UK.

### 8.2 Lag length selection for VAR model

For the proposed model, we build country-wise VAR representations with inflation, $\frac{APC \times (d-g) \times D}{GDP}$ and $\frac{(l-g) \times L}{GDP}$ as endogenous variables. Lag length in the range [1, 5] are tested. The lag length that minimizes different information criteria like LR, FPE, AIC, SC and HQ are noted. Lag numbers suggested by majority of the information criteria are selected. When there is a tie, we choose the minimum one. The lag length is thereby chosen to be: 4 (four) for Australia, 1 (one) for Japan & Korea and 2 (two) for Switzerland & UK. The summary of the lag order selection test for the proposed model is presented in Table 6.

After determining the appropriate lag length, we run our country-wise VAR model to check for the presence of serial correlation in the residuals. Serial Correlation LM Test is performed for lag length [1–10] and the results are presented in Table 7 for Australia, Table 8 for

**Table 1. ADF Unit Root Test.**

| Country | Series | Date Range | ADF Type | Level/ Differenced | p-value | Remark @ 5% |
|---------|--------|-----------|----------|--------------------|---------|-------------|
| Australia | Inflation | 1975-2013 | Intercept | L | 0.0805 | NS |
| | | ,, | | FD | 0 | S |
| | | ,, | Trend and Intercept | L | 0.1094 | NS |
| | | ,, | | FD | 0 | S |
| | | ,, | None | L | 0.0083 | S |
| | | ,, | | FD | 0 | S |
| | Nominal deposit rate | 1975-2013 | Intercept | L | 0.7614 | NS |
| | | ,, | | FD | 0.0002 | S |
| | | ,, | Trend and Intercept | L | 0.5587 | NS |
| | | ,, | | FD | 0.0013 | S |
| | | ,, | None | L | 0.2377 | NS |
| | | ,, | | FD | 0 | S |
| | Nominal lending rate | 1975-2013 | Intercept | L | 0.6253 | NS |
| | | ,, | | FD | 0 | S |
| | | ,, | Trend and Intercept | L | 0.4679 | NS |
| | | ,, | | FD | 0 | S |
| | | ,, | None | L | 0.3475 | NS |
| | | ,, | | FD | 0 | S |
| | APCx(d-g)xD/GDP | 1975-2013 | Intercept | L | 0.2795 | NS |
| | | ,, | | FD | 0 | S |
| | | ,, | Trend and Intercept | L | 0.253 | NS |
| | | ,, | | FD | 0 | S |
| | | ,, | None | L | 0.1008 | NS |
| | | ,, | | FD | 0 | S |
| | (l-g)xL/GDP | 1975-2013 | Intercept | L | 0.0438 | S |
| | | ,, | | FD | 0 | S |
| | | ,, | Trend and Intercept | L | 0.1293 | NS |
| | | ,, | | FD | 0 | S |
| | | ,, | None | L | 0.3454 | NS |
| | | ,, | | FD | 0 | S |

Japan, Table 9 for Korea, Table 10 for Switzerland and Table 11 for UK. From these tables, it is evident that none of the VAR models with lag length selected in the above manner suffers from the problem of serial correlation which is desirable.

We also check for the dynamic stability of the VAR models with selected lag length. It can be seen from Figs 5, 6, 7, 8 and 9 that all the models are dynamically stable (having their roots lying within the unit circle).

For the Fisherian model, we build country-wise VAR representations with inflation($t + 1$) and nominal lending rate($t$) as endogenous variables. The optimal lag length is selected to be: 1 (one) for Australia, Korea, Switzerland & UK and 2 (two) for Japan. The summary of the lag order selection test for the Fisherian model is presented in Table 12.

After determining the appropriate lag length, we run our country-wise VAR model to check for the presence of serial correlation in the residuals. Serial Correlation LM Test is performed for lag length [1–10] and the results are presented in Table 13 for Australia, Table 14 for Japan, Table 15 for Korea, Table 16 for Switzerland and Table 17 for UK. From these tables,

**Table 2. ADF Unit Root Test.**

| Country | Series | Date Range | ADF Type | Level/ Differenced | p-value | Remark @ 5% |
|---|---|---|---|---|---|---|
| Japan | Inflation | 1977-2013 | Intercept | L | 0.0232 | S |
| | | ,, | | FD | 0 | S |
| | | ,, | Trend and Intercept | L | 0.0187 | S |
| | | ,, | | FD | 0 | S |
| | | ,, | None | L | 0.0008 | S |
| | | ,, | | FD | 0 | S |
| | Nominal deposit rate | 1977-2013 | Intercept | L | 0.4554 | NS |
| | | ,, | | FD | 0.0003 | S |
| | | ,, | Trend and Intercept | L | 0.0545 | NS |
| | | ,, | | FD | 0.0018 | S |
| | | ,, | None | L | 0.1681 | NS |
| | | ,, | | FD | 0 | S |
| | Nominal lending rate | 1977-2013 | Intercept | L | 0.8317 | NS |
| | | ,, | | FD | 0.0002 | S |
| | | ,, | Trend and Intercept | L | 0.4698 | NS |
| | | ,, | | FD | 0.0014 | S |
| | | ,, | None | L | 0.0833 | NS |
| | | ,, | | FD | 0 | S |
| | APCx(d-g)xD/GDP | 1977-2013 | Intercept | L | 0.0001 | S |
| | | ,, | | FD | 0 | S |
| | | ,, | Trend and Intercept | L | 0.0004 | S |
| | | ,, | | FD | 0 | S |
| | | ,, | None | L | 0 | S |
| | | ,, | | FD | 0 | S |
| | (l-g)xL/GDP | 1977-2013 | Intercept | L | 0.0001 | S |
| | | ,, | | FD | 0 | S |
| | | ,, | Trend and Intercept | L | 0.0005 | S |
| | | ,, | | FD | 0 | S |
| | | ,, | None | L | 0.0003 | S |
| | | ,, | | FD | 0 | S |

it is evident that none of the VAR models with lag length selected in the above manner suffers from the problem of serial correlation which is desirable.

We then check for the dynamic stability of the VAR models with selected lag length. It can be seen from Figs 10, 11, 12, 13 and 14 that all the models are dynamically stable (having their roots lying within the unit circle).

## 8.3 VAR Granger Causality/Block Exogeneity Wald Test (T-Y approach)

Having determined the value of $m$ and $p$, we are now in the position to run the VAR Granger Causality/Block Exogeneity Wald Test. We insert inflation, $\frac{APC \times (d-g) \times D}{GDP}$ and $\frac{(l-g) \times L}{GDP}$ as endogenous variables in unrestricted VAR estimation while the lag number $p$ for the endogenous variables are already calculated in previous sections. We add additional $m$ lags of inflation, $\frac{APC \times (d-g) \times D}{GDP}$ and $\frac{(l-g) \times L}{GDP}$ as exogenous variables in the VAR. With this specification, we perform VAR Granger Causality/Block Exogeneity Wald Test on our data. The results of the test for our model are presented in Table 18. From Table 18, it is evident that we have found Granger Causality from two of our proposed metrics namely, $\frac{APC \times (d-g) \times D}{GDP}$ and $\frac{(l-g) \times L}{GDP}$ to inflation @1%

**Table 3. ADF Unit Root Test.**

| Country | Series | Date Range | ADF Type | Level/ Differenced | p-value | Remark @ 5% |
|---------|--------|-----------|----------|-------------------|---------|-------------|
| Korea | Inflation | 1980-2013 | Intercept | L | 0.0001 | S |
| | | ,, | | FD | 0.0008 | S |
| | | ,, | Trend and Intercept | L | 0.0007 | S |
| | | ,, | | FD | 0.0048 | S |
| | | ,, | None | L | 0 | S |
| | | ,, | | FD | 0 | S |
| | Nominal deposit rate | 1980-2013 | Intercept | L | 0.0225 | S |
| | | ,, | | FD | 0.0002 | S |
| | | ,, | Trend and Intercept | L | 0.0089 | S |
| | | ,, | | FD | 0.0014 | S |
| | | ,, | None | L | 0.0102 | S |
| | | ,, | | FD | 0 | S |
| | Nominal lending rate | 1980-2013 | Intercept | L | 0.0661 | NS |
| | | ,, | | FD | 0.0002 | S |
| | | ,, | Trend and Intercept | L | 0.0511 | NS |
| | | ,, | | FD | 0.0009 | S |
| | | ,, | None | L | 0.0301 | S |
| | | ,, | | FD | 0 | S |
| | APCx(d-g)xD/GDP | 1980-2013 | Intercept | L | 0 | S |
| | | ,, | | FD | 0 | S |
| | | ,, | Trend and Intercept | L | 0.0002 | S |
| | | ,, | | FD | 0 | S |
| | | ,, | None | L | 0 | S |
| | | ,, | | FD | 0 | S |
| | (l-g)xL/GDP | 1980-2013 | Intercept | L | 0.0001 | S |
| | | ,, | | FD | 0 | S |
| | | ,, | Trend and Intercept | L | 0.0002 | S |
| | | ,, | | FD | 0 | S |
| | | ,, | None | L | 0.0001 | S |
| | | ,, | | FD | 0 | S |

level for Australia, Japan, Korea and Switzerland. However, no causality is conferred by the test for the British data.

On the other hand, the results of performing VAR Granger Causality under Fisherian framework are presented in Tables 19 and 20. From Table 19, we find evidence in favour of Granger Causality running from expected inflation (actual inflation at time $(t + 1)$) to (current) nominal lending rate (nominal lending rate at time $t$). However, the causality in the opposite direction i.e., from nominal lending rate$(t)$ to inflation$(t + 1)$ does not hold true in any of the cases as depicted in Table 20.

## 8.4 ARDL Bounds Test

ARDL Bounds Testing approach proposed by Pesaran, Shin and Smith (2001) [13] can be performed on different parametric settings. For example, different kind of fixed regressors can be incorporated into the model: intercept, intercept and trend, no intercept no trend etc. We try all these three variants. We set the maximum lag for dependent variable and regressors to be 5. On these specification, ARDL Bounds Testing is performed.

**Table 4. ADF Unit Root Test.**

| Country | Series | Date Range | ADF Type | Level/ Differenced | p-value | Remark @ 5% |
|---|---|---|---|---|---|---|
| Switzerland | Inflation | 1981-2013 | Intercept | L | 0.1038 | S |
| | | ,, | | FD | 0 | S |
| | | ,, | Trend and Intercept | L | 0.0765 | NS |
| | | ,, | | FD | 0 | S |
| | | ,, | None | L | 0.0183 | S |
| | | ,, | | FD | 0 | S |
| | Nominal deposit rate | 1981-2013 | Intercept | L | 0.1477 | NS |
| | | ,, | | FD | 0.0002 | S |
| | | ,, | Trend and Intercept | L | 0.1127 | NS |
| | | ,, | | FD | 0.0019 | S |
| | | ,, | None | L | 0.0135 | S |
| | | ,, | | FD | 0 | S |
| | Nominal lending rate | 1981-2013 | Intercept | L | 0.5547 | NS |
| | | ,, | | FD | 0.0056 | S |
| | | ,, | Trend and Intercept | L | 0.3216 | NS |
| | | ,, | | FD | 0.0308 | S |
| | | ,, | None | L | 0.1722 | NS |
| | | ,, | | FD | 0.0004 | S |
| | APCx(d-g)xD/GDP | 1981-2013 | Intercept | L | 0.1654 | NS |
| | | ,, | | FD | 0 | S |
| | | ,, | Trend and Intercept | L | 0.1023 | NS |
| | | ,, | | FD | 0 | S |
| | | ,, | None | L | 0.014 | S |
| | | ,, | | FD | 0 | S |
| | (l-g)xL/GDP | 1981-2013 | Intercept | L | 0.0838 | NS |
| | | ,, | | FD | 0 | S |
| | | ,, | Trend and Intercept | L | 0.0942 | NS |
| | | ,, | | FD | 0.0001 | S |
| | | ,, | None | L | 0.0699 | NS |
| | | ,, | | FD | 0 | S |

**8.4.1 ARDL Bounds Testing under proposed framework.** For our model, we insert inflation as dependent variable and $\frac{APC \times (d-g) \times D}{GDP}$ & $\frac{(l-g) \times L}{GDP}$ as two dynamic regressors. The results are presented in Tables 21, 22, 23 and 24. From these tables, we can see the presence of long run relationships in Australian and Korean data for all three ARDL variants. For Japanese and Swiss data, we find the existence of long run relationship amongst the variables for 2 out of 3 variants of ARDL modelling.

Table 21 depicts the ARDL Bounds Testing result for Australian data under proposed framework. It can be seen from Table 21 that F-statistics of Bounds Test are found to be 6.155656, 17.13076 and 5.809287 which are greater than the corresponding I1 bound of 4.85, 5.85 and 3.83 respectively. Thus there are long run equilibrium relationships among the variables under all three variants of ARDL modelling. Moreover, the speed of adjustments for all three variants are found to be negative which implies that the process will converge to its long run equilibrium once distorted. P-values corresponding to the speed of adjustment are found to be significant even at 2% level for ARDL models with constant and linear trend as fixed regressor. However, p-value of speed of adjustment for ARDL models with no fixed regressor

**Table 5. ADF Unit Root Test.**

| Country | Series | Date Range | ADF Type | Level/ Differenced | p-value | Remark @ 5% |
|---|---|---|---|---|---|---|
| UK | Inflation | 1970-1998 | Intercept | L | 0.3424 | NS |
| | | ,, | | FD | 0.4762 | NS |
| | | ,, | | SD | 0 | S |
| | | ,, | Trend and Intercept | L | 0.6807 | NS |
| | | ,, | | FD | 0.7904 | NS |
| | | ,, | | SD | 0 | S |
| | | ,, | None | L | 0.0012 | S |
| | | ,, | | FD | 0.17 | NS |
| | | ,, | | SD | 0 | S |
| | Nominal deposit rate | 1970-1998 | Intercept | L | 0.2857 | NS |
| | | ,, | | FD | 0.0004 | S |
| | | ,, | Trend and Intercept | L | 0.5305 | NS |
| | | ,, | | FD | 0.0032 | S |
| | | ,, | None | L | 0.3704 | NS |
| | | ,, | | FD | 0 | S |
| | Nominal lending rate | 1970-1998 | Intercept | L | 0.0736 | NS |
| | | ,, | | FD | 0.0006 | S |
| | | ,, | Trend and Intercept | L | 0.1727 | NS |
| | | ,, | | FD | 0.0017 | S |
| | | ,, | None | L | 0.5273 | NS |
| | | ,, | | FD | 0 | S |
| | APCx(d-g)xD/GDP | 1970-1998 | Intercept | L | 0.0212 | S |
| | | ,, | | FD | 0.0039 | S |
| | | ,, | Trend and Intercept | L | 0.0886 | NS |
| | | ,, | | FD | 0.0142 | S |
| | | ,, | None | L | 0.0363 | S |
| | | ,, | | FD | 0.0002 | S |
| | (l-g)xL/GDP | 1970-1998 | Intercept | L | 0.0317 | S |
| | | ,, | | FD | 0.003 | S |
| | | ,, | Trend and Intercept | L | 0.0713 | NS |
| | | ,, | | FD | 0.013 | S |
| | | ,, | None | L | 0.2658 | NS |
| | | ,, | | FD | 0.0001 | S |

is not found to be significant at 5% level. Results are found to be stable as can be seen from the output of CUSUM test as depicted in Figs 15, 16 and 17.

Table 22 depicts the ARDL Bounds Testing result for Japanese data under proposed framework. It can be seen from Table 22 that F-statistics of Bounds Test are found to be 3.346849,

**Table 6. Lag length selection (proposed model).**

| Country | Time range | Max Lag | p [min LR] | p [min FPE] | p [min AIC] | p [min SC] | p [min HQ] |
|---|---|---|---|---|---|---|---|
| Australia | 1975-2013 | 5 | 4 | 4 | 5 | 2 | 4 |
| Japan | 1977-2013 | 5 | 1 | 1 | 1 | 1 | 1 |
| Korea | 1980-2013 | 5 | 1 | 1 | 1 | 1 | 1 |
| Switzerland | 1981-2013 | 5 | 2 | 2 | 2 | 2 | 2 |
| UK | 1970-1998 | 5 | 1 | 2 | 2 | 1 | 2 |

**Table 7. Serial correlation LM test for Australian data when P = 4 (proposed model).**

| Lags | LM-Stat | Prob |
|---|---|---|
| 1 | 7.579139 | 0.5771 |
| 2 | 7.692969 | 0.5654 |
| 3 | 9.598352 | 0.384 |
| 4 | 7.814252 | 0.553 |
| 5 | 7.925114 | 0.5417 |
| 6 | 15.90506 | 0.0689 |
| 7 | 16.50035 | 0.0571 |
| 8 | 11.25349 | 0.2587 |
| 9 | 9.900822 | 0.3586 |
| 10 | 8.496971 | 0.4849 |

**Table 8. Serial correlation LM test for Japanese data when P = 1 (proposed model).**

| Lags | LM-Stat | Prob |
|---|---|---|
| 1 | 13.56717 | 0.1386 |
| 2 | 16.97337 | 0.0491 |
| 3 | 2.601459 | 0.978 |
| 4 | 3.873359 | 0.9195 |
| 5 | 3.502116 | 0.941 |
| 6 | 18.13745 | 0.0336 |
| 7 | 6.485064 | 0.6906 |
| 8 | 3.056369 | 0.962 |
| 9 | 8.485173 | 0.4861 |
| 10 | 3.423177 | 0.9451 |

10.19052 and 5.185616 and the corresponding I1 bounds are found to be 4.85, 5.85 and 3.83 respectively. This implies that no long run relationship exists among the variables modeled under ARDL framework with constant fixed regressor. However, for ARDL with linear trend and ARDL with no fixed regressor confer the existence of long run equilibrating relationships among the variables. Moreover, the speed of adjustment for ARDL with linear trend is found to be negative and significant even at 1% level. For ARDL with no fixed regressor, the speed of

**Table 9. Serial correlation LM test for Korean data when P = 1 (proposed model).**

| Lags | LM-Stat | Prob |
|---|---|---|
| 1 | 7.258296 | 0.6102 |
| 2 | 7.983905 | 0.5358 |
| 3 | 4.182982 | 0.899 |
| 4 | 3.797664 | 0.9242 |
| 5 | 7.003602 | 0.6367 |
| 6 | 2.864117 | 0.9694 |
| 7 | 5.647841 | 0.7746 |
| 8 | 12.00255 | 0.2132 |
| 9 | 10.33387 | 0.3241 |
| 10 | 2.735622 | 0.9739 |

**Table 10. Serial correlation LM test for Swiss data when P = 2 (proposed model).**

| Lags | LM-Stat | Prob |
|---|---|---|
| 1 | 18.03033 | 0.0348 |
| 2 | 7.984452 | 0.5357 |
| 3 | 13.04658 | 0.1605 |
| 4 | 6.731945 | 0.665 |
| 5 | 6.747347 | 0.6634 |
| 6 | 6.531909 | 0.6857 |
| 7 | 7.680773 | 0.5666 |
| 8 | 7.815361 | 0.5529 |
| 9 | 7.452108 | 0.5902 |
| 10 | 3.901727 | 0.9178 |

adjustment is found to be -0.114194 which is desirable. However, the corresponding p-value is not found to be significant at 2% level as can be seen from Table 22. Moreover, the long run cointegrating relationships are found to be stable as can be seen from the outcome of CUSUM tests as depicted in Figs 18 and 19.

Table 23 presents the ARDL Bounds Testing result of Korean data under proposed framework. Here, for all three variants of ARDL modelling, F-statistics are found to be significantly greater than the corresponding I1 bounds. Moreover, in all cases, the speeds of adjustments are found to be negative and significant even at 1% level. Last but not the least, the results are found to be stable in all cases except ARDL with no fixed regressor as can be seen from the output of CUSUM test as depicted in Figs 20, 21 and 22.

Table 24 presents the ARDL Bounds Testing results for Swiss data under proposed framework. As evident from Table 24, ARDL models with constant fixed regressor and linear trend entail long run equilibrating relationships among the variables. However, no long run relationship is exhibited in ARDL models with no fixed regressor. Moreover, speeds of adjustments are found to be negative and significant at 1% level and the results are found to dynamically stable (as can be seen from Figs 23 and 24).

**8.4.2 ARDL Bounds Testing under Fisherian framework.** ARDL Bounds Testing under Fisherian framework is performed with nominal lending rate($t$) as dependent variable and inflation($t + 1$) as independent variable. Maximum lag length for dependent variable and dynamic regressors are chosen to be 5 as before. All three variants with different kinds of fixed

**Table 11. Serial correlation LM test for British data when P = 2 (proposed model).**

| Lags | LM-Stat | Prob |
|---|---|---|
| 1 | 10.74953 | 0.2933 |
| 2 | 9.332814 | 0.4071 |
| 3 | 5.886589 | 0.7512 |
| 4 | 5.679308 | 0.7715 |
| 5 | 4.020521 | 0.9101 |
| 6 | 4.642957 | 0.8643 |
| 7 | 7.417018 | 0.5938 |
| 8 | 9.108675 | 0.4273 |
| 9 | 4.024058 | 0.9098 |
| 10 | 7.625389 | 0.5723 |

## Inverse Roots of AR Characteristic Polynomial

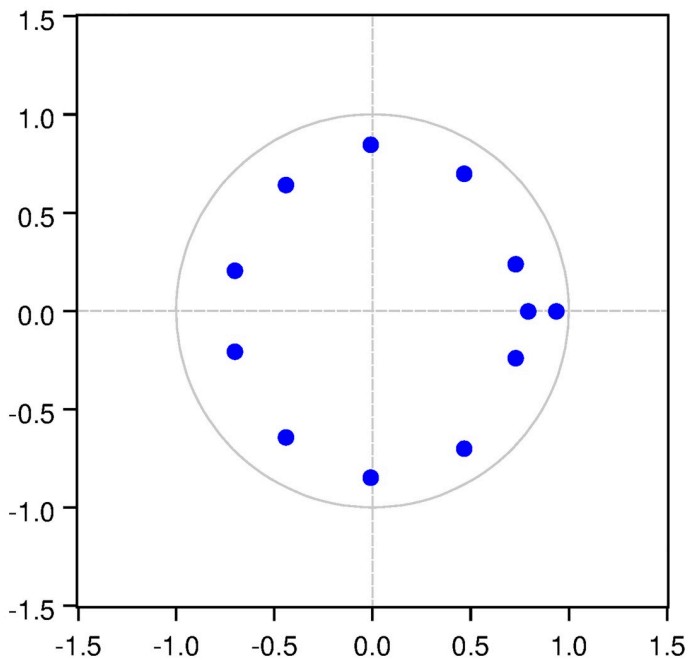

**Fig 5. Inverse roots of AR characteristic polynomial for Australian data when P = 4 (proposed model).**

## Inverse Roots of AR Characteristic Polynomial

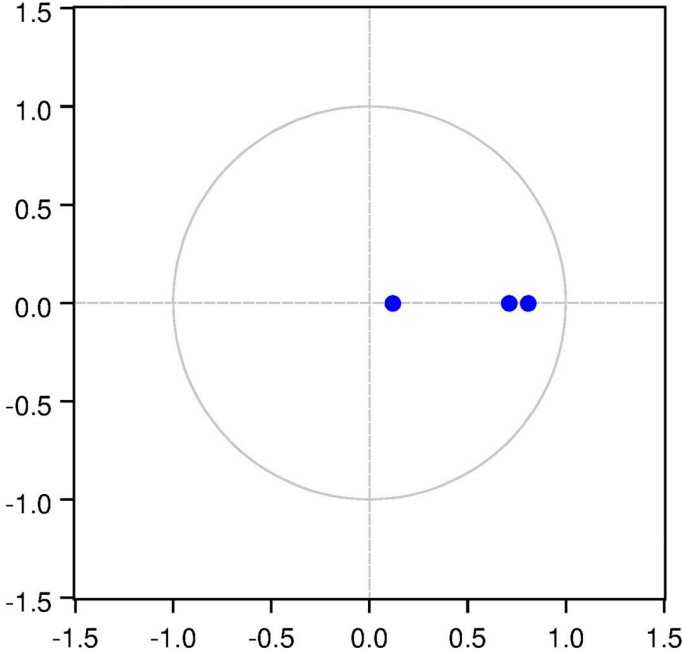

**Fig 6. Inverse roots of AR characteristic polynomial for Japanese data when P = 1 (proposed model).**

## Inverse Roots of AR Characteristic Polynomial

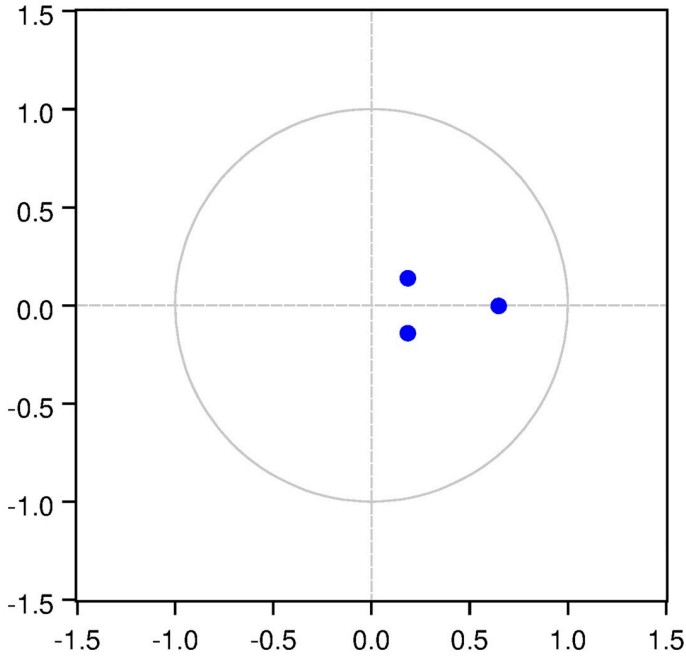

**Fig 7. Inverse roots of AR characteristic polynomial for Korean data when P = 1 (proposed model).**

## Inverse Roots of AR Characteristic Polynomial

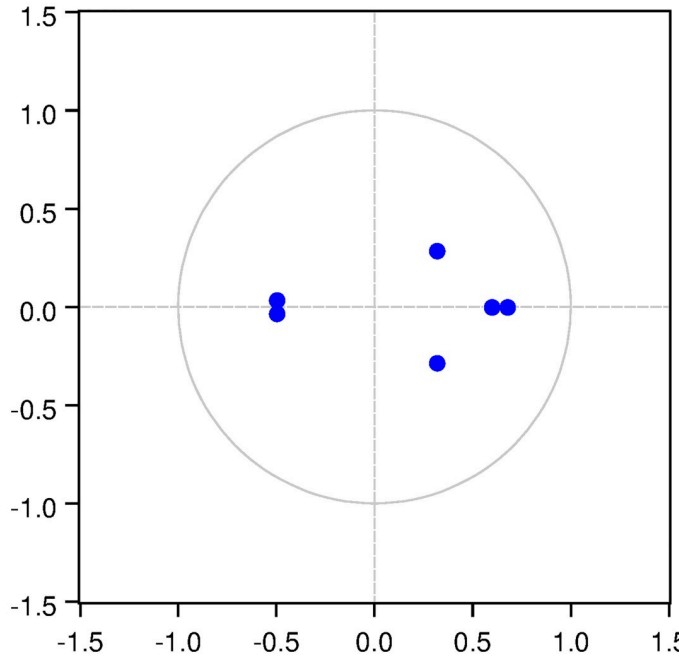

**Fig 8. Inverse roots of AR characteristic polynomial for Swiss data when P = 2 (proposed model).**

## Inverse Roots of AR Characteristic Polynomial

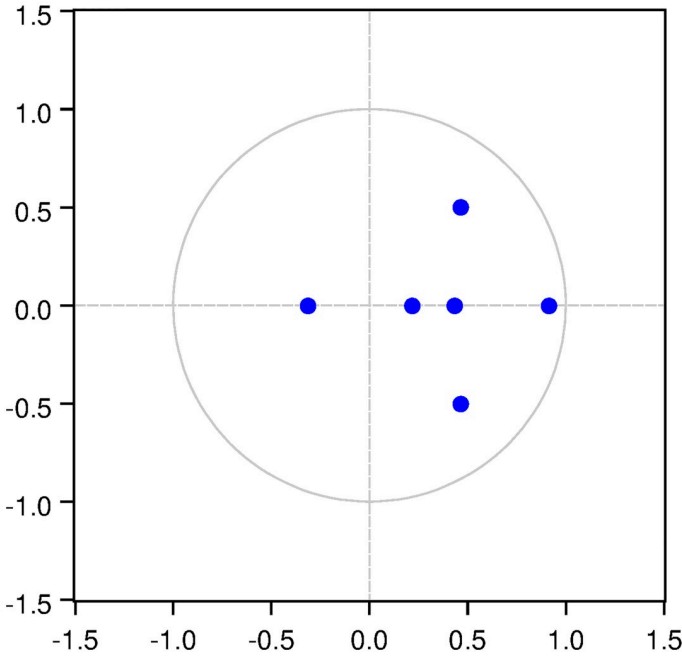

**Fig 9. Inverse roots of AR characteristics polynomial for British data when P = 2 (proposed model).**

**Table 12. Lag length selection (Fisherian model).**

| Country | Time range | Max Lag | p [min LR] | p [min FPE] | p [min AIC] | p [min SC] | p [min HQ] |
|---|---|---|---|---|---|---|---|
| Australia | 1975-2013 | 5 | 3 | 5 | 5 | 1 | 1 |
| Japan | 1977-2013 | 5 | 5 | 2 | 5 | 1 | 2 |
| Korea | 1980-2013 | 5 | 1 | 1 | 1 | 1 | 1 |
| Switzerland | 1981-2013 | 5 | 1 | 1 | 1 | 1 | 1 |
| UK | 1970-1998 | 5 | 1 | 3 | 3 | 1 | 1 |

**Table 13. Serial correlation LM test for Australian data when P = 1 (Fisherian model).**

| Lags | LM-Stat | Prob |
|---|---|---|
| 1 | 3.904531 | 0.4191 |
| 2 | 11.02384 | 0.0263 |
| 3 | 1.698263 | 0.791 |
| 4 | 10.34965 | 0.0349 |
| 5 | 2.402226 | 0.6622 |
| 6 | 4.183825 | 0.3817 |
| 7 | 1.812245 | 0.7702 |
| 8 | 3.530322 | 0.4733 |
| 9 | 2.554443 | 0.6349 |
| 10 | 1.840608 | 0.765 |

**Table 14. Serial correlation LM test for Japanese data when P = 2 (Fisherian model).**

| Lags | LM-Stat | Prob |
|---|---|---|
| 1 | 17.04802 | 0.0019 |
| 2 | 3.793051 | 0.4347 |
| 3 | 2.516511 | 0.6417 |
| 4 | 2.223534 | 0.6947 |
| 5 | 5.986499 | 0.2002 |
| 6 | 5.760069 | 0.2178 |
| 7 | 1.054205 | 0.9015 |
| 8 | 2.743471 | 0.6016 |
| 9 | 3.63647 | 0.4574 |
| 10 | 1.831837 | 0.7667 |

**Table 15. Serial correlation LM test for Korean data when P = 1 (Fisherian model).**

| Lags | LM-Stat | Prob |
|---|---|---|
| 1 | 9.898649 | 0.0422 |
| 2 | 3.595524 | 0.4635 |
| 3 | 3.274675 | 0.513 |
| 4 | 2.716095 | 0.6064 |
| 5 | 2.152406 | 0.7078 |
| 6 | 1.362994 | 0.8506 |
| 7 | 7.61194 | 0.1069 |
| 8 | 4.757626 | 0.3131 |
| 9 | 10.35274 | 0.0349 |
| 10 | 0.587677 | 0.9644 |

regressors are tested. The results are presented in Tables 25, 26, 27 and 28. We find evidences of existence of long run Fisher effect in all the countries under ARDL with no fixed regressor. In the other two variants of ARDL, no cointegrating relationship is observed for any country in the list.

For Australian data, F-statistic of Bounds Testing for ARDL with no fixed regressor is found to be 4.823703 which is higher than the corresponding I1 bound of 4.11. Moreover,

**Table 16. Serial correlation LM test for Swiss data when P = 1 (Fisherian model).**

| Lags | LM-Stat | Prob |
|---|---|---|
| 1 | 4.831434 | 0.305 |
| 2 | 7.917815 | 0.0946 |
| 3 | 0.827042 | 0.9348 |
| 4 | 3.456057 | 0.4846 |
| 5 | 1.131897 | 0.8892 |
| 6 | 4.763078 | 0.3125 |
| 7 | 4.528915 | 0.3391 |
| 8 | 7.925324 | 0.0944 |
| 9 | 0.382637 | 0.9839 |
| 10 | 4.806577 | 0.3077 |

**Table 17. Serial correlation LM test for British data when P = 1 (Fisherian model).**

| Lags | LM-Stat | Prob |
|---|---|---|
| 1 | 2.874381 | 0.5791 |
| 2 | 3.893729 | 0.4206 |
| 3 | 1.916448 | 0.7511 |
| 4 | 4.494705 | 0.3432 |
| 5 | 4.824935 | 0.3057 |
| 6 | 9.797158 | 0.044 |
| 7 | 1.708896 | 0.7891 |
| 8 | 3.464464 | 0.4833 |
| 9 | 1.616533 | 0.8058 |
| 10 | 3.829458 | 0.4296 |

speed of adjustment is found to be -0.152443 which is significant at 1% level (see Table 25). As evident from Fig 25, the model is found to be stable.

For Japanese data, F-statistic of Bounds Testing under Fisherian framework for ARDL with no fixed regressor is found to be 5.324548 whereas the corresponding I1 bound at 5% level is 4.11 which suggests the presence of long run cointegrating relationship among the variables (see Table 26). Speed of adjustment is -0.059149 which is negative and signifies that the process will eventually converge to its long run equilibrium once distorted. Moreover, the p-value corresponding to the speed of adjustment is found to be significant even at 1% level and the model is found to be stable dynamically as can be seen from Fig 26.

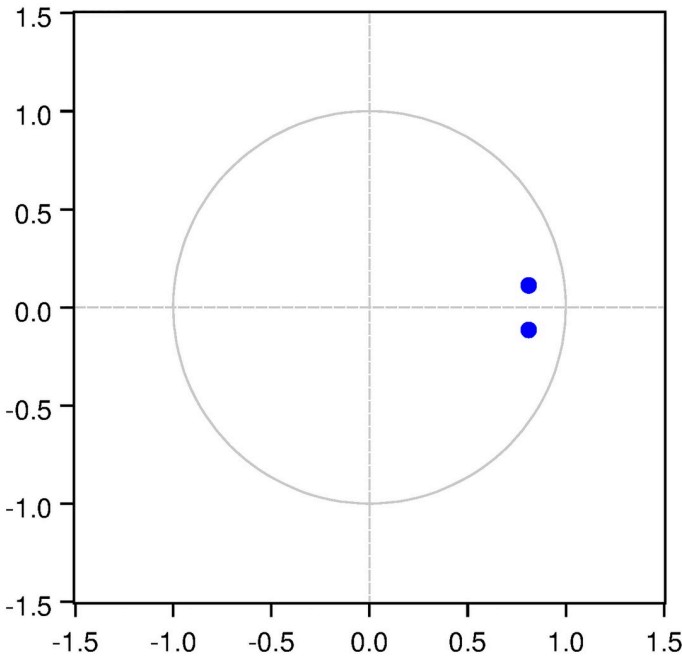

**Fig 10. Inverse roots of AR characteristic polynomial for Australian data when P = 1 (fisherian model).**

## Inverse Roots of AR Characteristic Polynomial

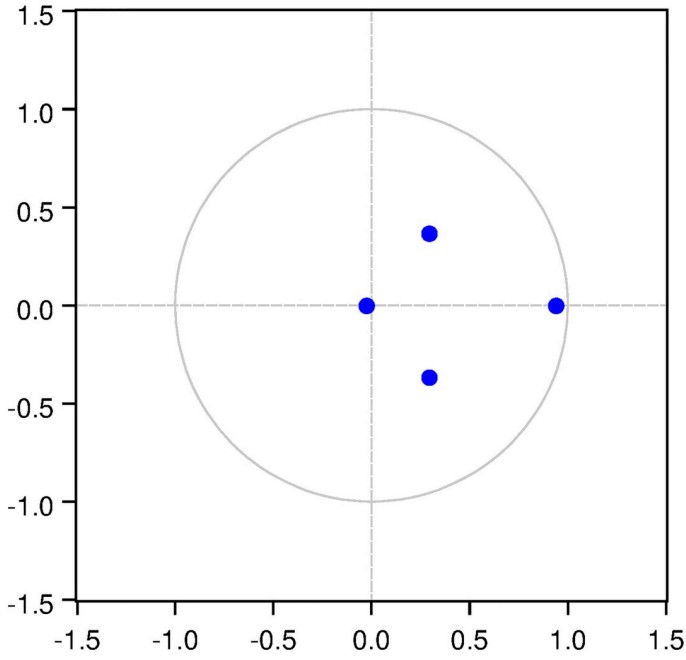

**Fig 11. Inverse roots of AR characteristic polynomial for Japanese data when P = 2 (Fisherian model).**

## Inverse Roots of AR Characteristic Polynomial

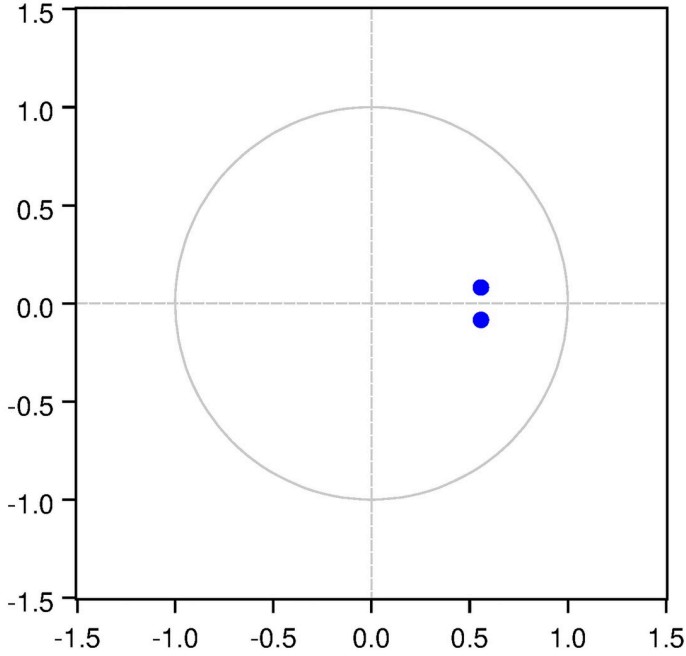

**Fig 12. Inverse roots of AR characteristic polynomial for Korean data when P = 1 (Fisherian model).**

## Inverse Roots of AR Characteristic Polynomial

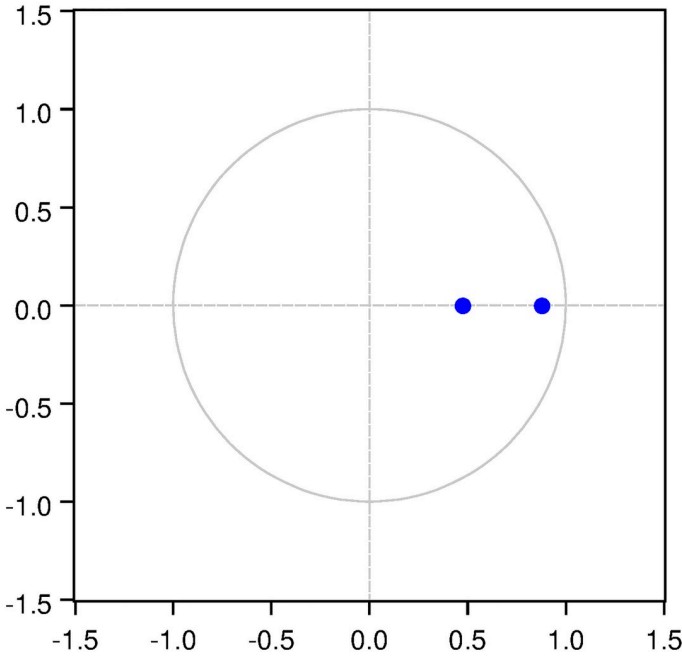

**Fig 13. Inverse roots of AR characteristic polynomial for Swiss data when P = 1 (Fisherian model).**

## Inverse Roots of AR Characteristic Polynomial

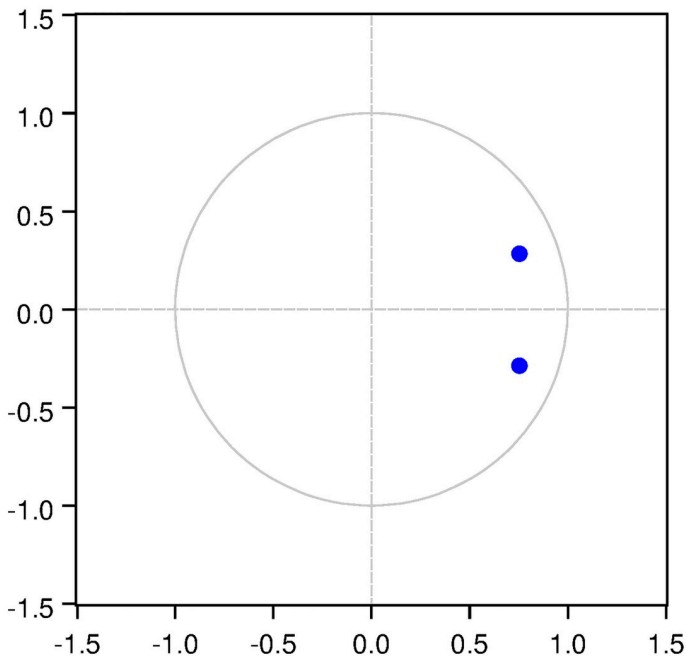

**Fig 14. Inverse roots of AR characteristic polynomial for British data when P = 1 (Fisherian model).**

**Table 18. VAR Granger Causality/Block Exogeneity Wald Test (proposed model).**

| Country | Time Range | m | p | Dependent variable | APCx(d-g)xD/GDP | (l-g)xL/GDP | Chi-Sq | df | p-value | Remark |
|---------|-----------|---|---|--------------------|------------------|-------------|--------|----|---------|--------|
| Australia | 1975-2013 | 1 | 4 | Inflation | excluded | excluded | 27.62508 | 8 | 0.0006 | Causality @1% |
| Japan | 1977-2013 | 0 | 1 | Inflation | excluded | excluded | 9.226999 | 2 | 0.0099 | Causality @1% |
| Korea | 1980-2013 | 0 | 1 | Inflation | excluded | excluded | 12.17776 | 2 | 0.0023 | Causality @1% |
| Switzerland | 1981-2013 | 1 | 2 | Inflation | excluded | excluded | 20.76112 | 4 | 0.0004 | Causality @1% |
| UK | 1970-1998 | 2 | 2 | Inflation | excluded | excluded | 6.529262 | 4 | 0.163 | No Causality |

**Table 19. VAR Granger Causality/Block Exogeneity Wald Test (Fisherian model).**

| Country | Time Range | m | p | Dependent variable(t) | Inflation (t+1) | Chi-Sq | df | p-value | Remark |
|---------|-----------|---|---|----------------------|-----------------|--------|----|---------|--------|
| Australia | 1975-2013 | 1 | 1 | Nominal lending rate | excluded | 3.638865 | 1 | 0.0564 | Causality @ 10% |
| Japan | 1977-2013 | 1 | 2 | Nominal lending rate | excluded | 18.57663 | 2 | 0.0001 | Causality @ 1% |
| Korea | 1980-2013 | 1 | 1 | Nominal lending rate | excluded | 8.830656 | 1 | 0.003 | Causality @ 1% |
| Switzerland | 1981-2013 | 1 | 1 | Nominal lending rate | excluded | 13.35468 | 1 | 0.0003 | Causality @ 1% |
| UK | 1970-1998 | 2 | 1 | Nominal lending rate | excluded | 2.826149 | 1 | 0.0927 | Causality @ 10% |

**Table 20. VAR Granger Causality/Block Exogeneity Wald Test (Fisherian model).**

| Country | Time Range | m | p | Dependent variable(t+1) | Nominal lending rate(t) | Chi-Sq | df | p-value | Remark |
|---------|-----------|---|---|------------------------|-------------------------|--------|----|---------|--------|
| Australia | 1975-2013 | 1 | 1 | Inflation | excluded | 1.423563 | 1 | 0.2328 | No Causality |
| Japan | 1977-2013 | 1 | 2 | Inflation | excluded | 0.593624 | 2 | 0.7432 | No Causality |
| Korea | 1980-2013 | 1 | 1 | Inflation | excluded | 0.052633 | 1 | 0.8185 | No Causality |
| Switzerland | 1981-2013 | 1 | 1 | Inflation | excluded | 1.642179 | 1 | 0.2 | No Causality |
| UK | 1970-1998 | 2 | 1 | Inflation | excluded | 0.927286 | 1 | 0.3356 | No Causality |

**Table 21. ARDL Bounds Testing for Australian data under proposed framework.**

| Country | Australia | | |
|---------|-----------|---|---|
| Date Range | 1975-2013 | | |
| Dependent Variable | Inflation | | |
| Independent Variable-1 | APCx(d-g)xM2/GDP | | |
| Independent Variable-2 | (l-g)xL/GDP | | |
| Dependent Variable: Max Lag | 5 | | |
| Regressor: Max Lag | 5 | | |
| Fixed Regressors | Constant | Linear Trend | None |
| Selected Model | (5, 5, 5) | (5, 3, 5) | (4, 4, 4) |
| F-Stat (Bound Test) | 6.155656 | 17.13076 | 5.809287 |
| I0 Bound (@5%) | 3.79 | 4.87 | 2.72 |
| I1 Bound (@ 5%) | 4.85 | 5.85 | 3.83 |
| Remark | Long run relationship exists | Long run relationship exists | Long run relationship exists |
| Speed of Adjustment | -0.350514 | -0.573077 | -0.061808 |
| p-value | 0.0179 | 0.0002 | 0.4052 |

**Table 22. ARDL Bounds Testing for Japanese data under proposed framework.**

| Country | Japan | | |
|---|---|---|---|
| Date Range | 1977-2013 | | |
| Dependent Variable | Inflation | | |
| Independent Variable-1 | APCx(d-g)xM2/GDP | | |
| Independent Variable-2 | (l-g)xL/GDP | | |
| Dependent Variable: Max Lag | 5 | | |
| Regressor: Max Lag | 5 | | |
| Fixed Regressors | Constant | Linear Trend | None |
| Selected Model | (3, 2, 2) | (1, 0, 2) | (3, 2, 2) |
| F-Stat (Bound Test) | 3.346849 | 10.19052 | 5.185616 |
| I0 Bound (@5%) | 3.79 | 4.87 | 2.72 |
| I1 Bound (@ 5%) | 4.85 | 5.85 | 3.83 |
| Remark | No long run relationship | Long run relationship exists | Long run relationship exists |
| Speed of Adjustment | - | -0.611663 | -0.114194 |
| p-value | - | 0.0016 | 0.1467 |

Table 27 presents the ARDL Bounds Testing result for Korean data under Fisherian framework. From Table 27, it is evident that the F-statistic for the ARDL model with no fixed regressor is considerably higher than the corresponding I1 bound at 5% level. Moreover, the speed of adjustment is found to be -0.17959 which is desirable and significant at 1% level. Last but not least, the CUSUM test result suggests the dynamic stability of the model (See Fig 27 for the result of CUSUM test).

Table 28 depicts the ARDL Bounds Testing result for Swiss data. Like all other cases reported above, we find long run cointegrating relationship among the variables only for the ARDL model with no fixed regressor. Here, the F-statistic of ARDL Bounds Test is 4.456717 which is greater than the corersponding I1 bound of 4.11. Moreover, the speed of adjustment is negative and significant at 2% level (see Table 28 for details). Lastly, the CUSUM test confers the stability of the model (as evident from Fig 28).

**Table 23. ARDL Bounds Testing for Korean data under proposed framework.**

| Country | Korea | | |
|---|---|---|---|
| Date Range | 1980-2013 | | |
| Dependent Variable | Inflation | | |
| Independent Variable-1 | APCx(d-g)xM2/GDP | | |
| Independent Variable-2 | (l-g)xL/GDP | | |
| Dependent Variable: Max Lag | 5 | | |
| Regressor: Max Lag | 5 | | |
| Fixed Regressors | Constant | Linear Trend | None |
| Selected Model | (1, 1, 1) | (1, 0, 1) | (1, 0, 1) |
| F-Stat (Bound Test) | 10.29754 | 10.60872 | 6.816165 |
| I0 Bound (@5%) | 3.79 | 4.87 | 2.72 |
| I1 Bound (@ 5%) | 4.85 | 5.85 | 3.83 |
| Remark | Long run relationship exists | Long run relationship exists | Long run relationship exists |
| Speed of Adjustment | -0.57676 | -0.47282 | -0.228168 |
| p-value | 0 | 0.0002 | 0.002 |

**Table 24. ARDL Bounds Testing for Swiss data under proposed framework.**

| Country | Switzerland | | |
|---|---|---|---|
| Date Range | 1981-2013 | | |
| Dependent Variable | Inflation | | |
| Independent Variable-1 | APCx(d-g)xM2/GDP | | |
| Independent Variable-2 | (l-g)xL/GDP | | |
| Dependent Variable: Max Lag | 5 | | |
| Regressor: Max Lag | 5 | | |
| Fixed Regressors | Constant | Linear Trend | None |
| Selected Model | (1, 3, 4) | (1, 4, 5) | (3, 5, 4) |
| F-Stat (Bound Test) | 10.96167 | 10.65164 | 0.346987 |
| I0 Bound (@5%) | 3.79 | 4.87 | 2.72 |
| I1 Bound (@ 5%) | 4.85 | 5.85 | 3.83 |
| Remark | Long run relationship exists | Long run relationship exists | No long run relationship |
| Speed of Adjustment | -1.167516 | -1.467213 | - |
| p-value | 0 | 0.0001 | - |

## 9 Discussion

If two or more time series are cointegrated then there is supposed to be Granger Causality amongst them in either direction or both. Results obtained here mostly agree with the above statement. To be precise, both of our variables namely $\frac{APC \times (d-g) \times D}{GDP}$ and $\frac{(l-g) \times L}{GDP}$ are found to be cointegrated with inflation for 4 out of 5 countries (as can be seen from Tables 21, 22, 23 and 24). For UK, we can not run the ARDL Bounds Test as one of the variables namely inflation is found to be non-stationary even after first difference (see Table 5) which invalidates the test. For the remaining four countires, cointegration amongst the proposed variables has been found. As cointegrations amongst the variables are found then we might assume the presence of Granger causality amongst the variables in at least one direction if not both. The presence of

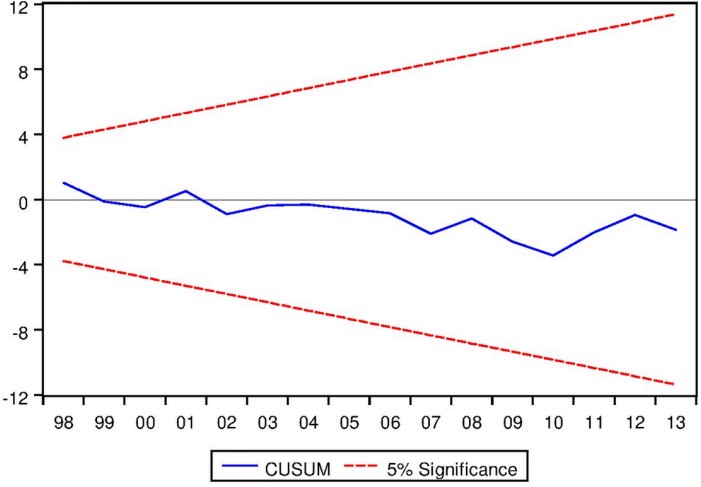

**Fig 15. Stability diagnostic of cointegrating relationship for Australian data with constand fixed regressor under proposed framework.**

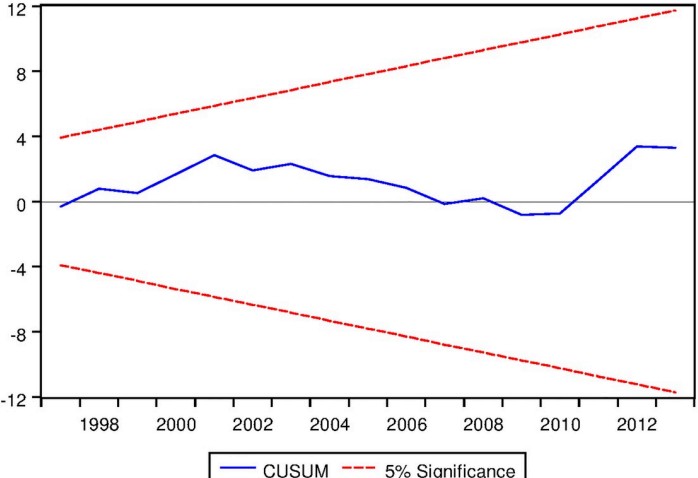

**Fig 16. Stability diagnostic of cointegrating relationship for Australian data with linear trend as fixed regressor under proposed framework.**

Granger causality from $\frac{APC \times (d-g) \times D}{GDP}$ and $\frac{(l-g) \times L}{GDP}$ to inflation for all the countries except UK has also been observed (as can be seen from Table 18) which reinforces our claim.

On the other hand, Fisher equation being an equality posits the presence of a bi-directional causality running between interest rate and inflation. As can be seen from Table 19, the Fisher equation can successfully explain the causal relationship running from expected (future) inflation to the (current) nominal lending rate. However, no causality is conferred in the reverse direction (see Table 20). So, although, inflation alone can explain interest rate, the converse is not necessarily true which implies it is better to view the Fisher effect as a unidirectional causality instead of a (bidirectional) equality. Infact, apart from interest rate, we need more variables to explain inflation and this is where lies the main essence of this paper. Here we argue

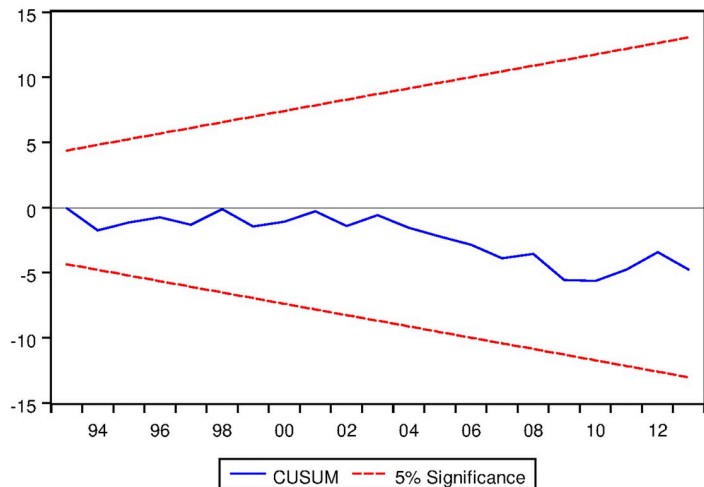

**Fig 17. Stability diagnostic of cointegrating relationship for Australian data with no fixed regressor under proposed framework.**

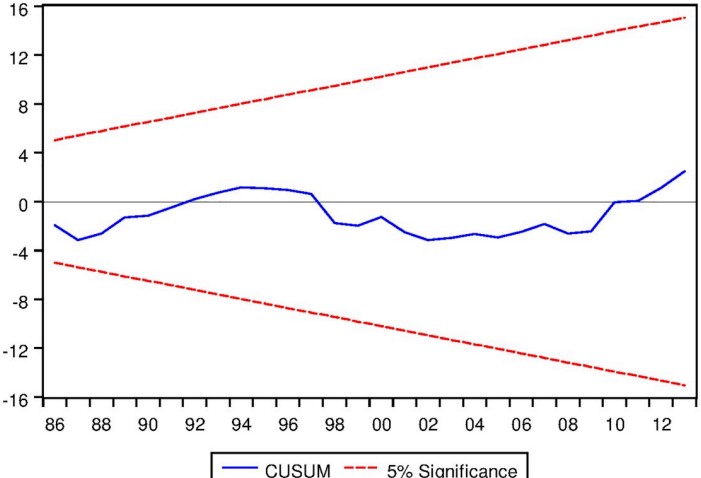

**Fig 18. Stability diagnostic of cointegrating relationship for Japanese data with linear trend as fixed regressor under proposed framework.**

interest rate when combined with real GDP growth rate, total amount of domestic credit and the total volume of deposit in the aforementioned manner can explain inflation. The empirical evidence in 4(four) out of 5(five) countries also suggests our intuitive arguments as can be seen from Table 18.

## 10 Conclusion

We compare the performance of our model with the Fisherian one using VAR Granger Causality Test and ARDL Bounds Test. This comparison is indeed necessary to provide a justification about why we should rethink the relationship between interest rate and inflation in greater detail above and beyond the Fisher equation. Fisher equation seeks to establish a

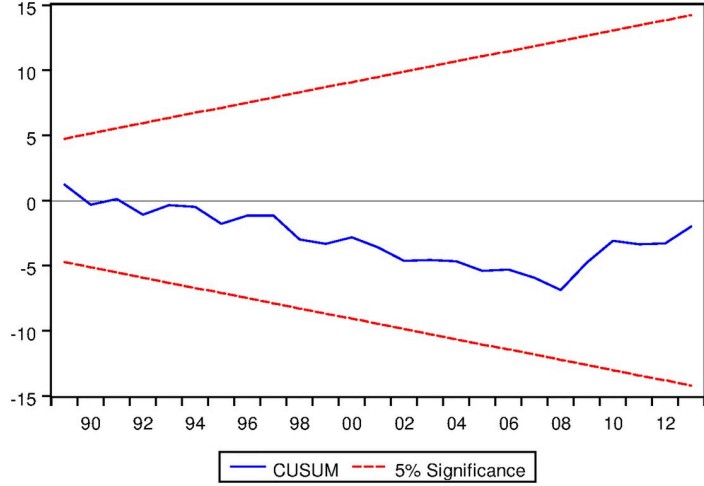

**Fig 19. Stability diagnostic of cointegrating relationship for Japanese data with no fixed regressor under proposed framework.**

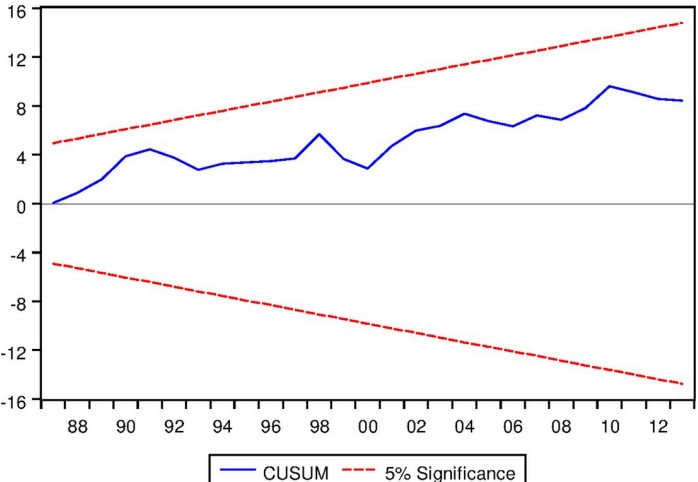

**Fig 20. Stability diagnostic of cointegrating relationship for Korean data with constant fixed regressor under proposed framework.**

relationship between interest rate and inflation based upon a causality which runs from expected inflation (future inflation) to the (current) nominal lending rate. Intuitively, when the lender anticipates a rise in inflation, he/she will set the nominal lending rate to a relatively higher value in order to compensate for the loss of purchasing power of money due to inflation. This is one angle from which we can see the dynamic relationship between interest rate and inflation. However, in this paper, we view the relationship from an angle different from the Fisherian one. In our proposed model, the causality goes from interest rate to inflation. Here, we argue that a change in nominal interest rate, if not accompanied by the same change in real GDP growth rate, can give birth to inflation. In almost all of the cases, the statistical analysis suggests long run (causal) relationship between the two proposed metrics and inflation. However, for a single case, we fail to find a causal relationship in our proposed direction.

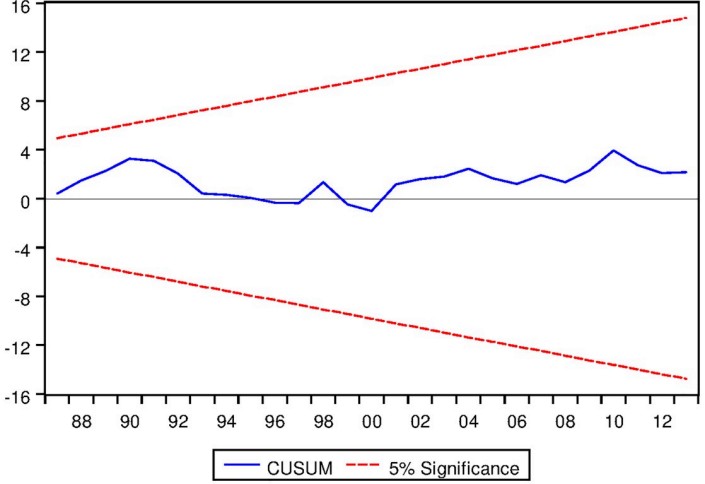

**Fig 21. Stability diagnostic of cointegrating relationship for Korean data with linear trend as fixed regressor under proposed framework.**

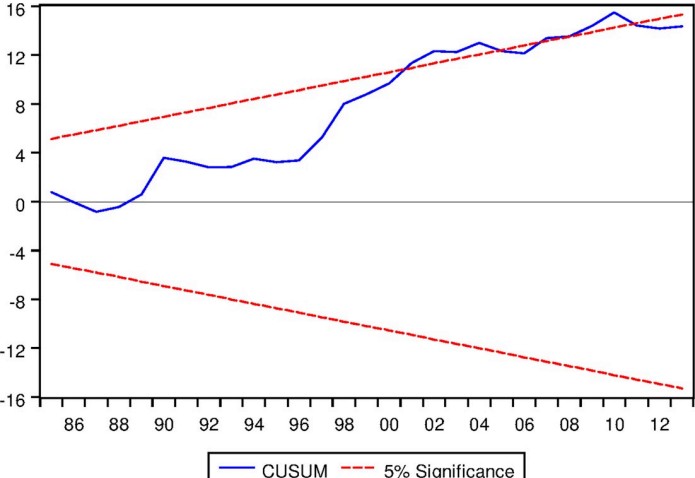

**Fig 22. Stability diagnostic of cointegrating relationship for Korean data with no fixed regressor under proposed framework.**

It is because, we have only considered a hand full of variables (two types of interest rate, total volume of deposit & credit in the banking system and the real GDP growth rate) to explain inflation. There is a whole set of other macro-economic phenomena which can influence inflation significantly. When the effect of the two proposed metrics are suppressed by the effect of some other phenomena acting on inflation in the opposite direction, then we think, we fail to find any significant cointegrating relationship and these deviations require detailed case-by-case analysis for every individual incident which is beyond the scope of this study. Yet, these two metrics can be used to explain inflation in the long run under broad head.

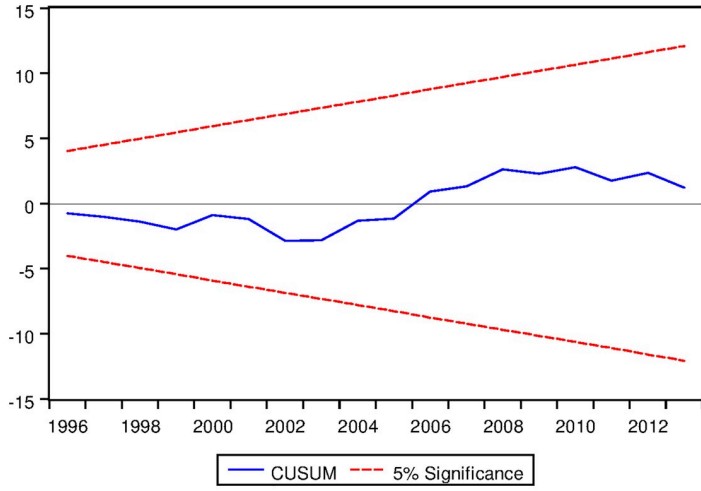

**Fig 23. Stability diagnostic of cointegrating relationship for Swiss data with constant fixed regressor under proposed framework.**

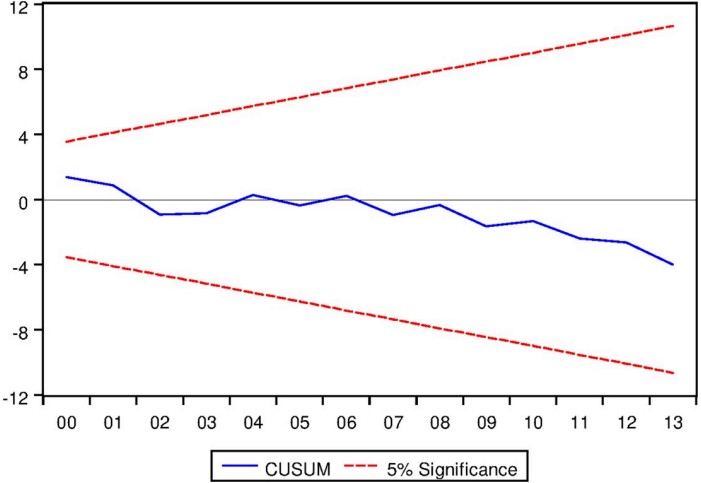

**Fig 24. Stability diagnostic of cointegrating relationship for Swiss data with linear trend as fixed regressor under proposed framework.**

# 11 Algorithms, figures and tables

## 11.1 Steps followed for Toda-Yamamoto approach of testing Granger Causality in the context of non-stationary time series

1. Determine the maximum order of integration of the underlying time series. Let this be denoted by $m$.

2. Determine the appropriate lag length for the VAR model having the data in level using some information criterion like LR, FPE, AIC, SC, HQ etc. The lag length that minimizes the chosen information criterion is selected.

3. Build a VAR model using all the endogenous variables in level each having number of lags as determined in the previous step.

**Table 25. ARDL Bounds Testing for Australian data under Fisherian framework.**

| Country | Australia | | |
|---|---|---|---|
| Date Range | 1975-2013 | | |
| Dependent Variable | Lending rate(t) | | |
| Independent Variable | Inflation(t+1) | | |
| Dependent Variable: Max Lag | 5 | | |
| Regressor: Max Lag | 5 | | |
| Fixed Regressors | Constant | Linear Trend | None |
| Selected Model | (3, 0) | (4, 5) | (3, 0) |
| F-Stat (Bound Test) | 4.562675 | 6.361843 | 4.823703 |
| I0 Bound (@5%) | 4.94 | 6.56 | 3.15 |
| I1 Bound (@ 5%) | 5.73 | 7.3 | 4.11 |
| Remark | No long run relationship | No long run relationship | Long run relationship exists |
| Speed of adjustment | - | - | -0.152443 |
| p-value | - | - | 0.0002 |

**Table 26. ARDL Bounds Testing for Japanese data under Fisherian framework.**

| Country | Japan | | |
|---|---|---|---|
| Date Range | 1977-2013 | | |
| Dependent Variable | Lending rate(t) | | |
| Independent Variable | Inflation(t+1) | | |
| Dependent Variable: Max Lag | 5 | | |
| Regressor: Max Lag | 5 | | |
| Fixed Regressors | Constant | Linear Trend | None |
| Selected Model | (4, 5) | (4, 5) | (3, 0) |
| F-Stat (Bound Test) | 3.020623 | 4.391565 | 5.324548 |
| I0 Bound (@5%) | 4.94 | 6.56 | 3.15 |
| I1 Bound (@ 5%) | 5.73 | 7.3 | 4.11 |
| Remark | No long run relationship | No long run relationship | Long run relationship exists |
| Speed of adjustment | - | - | -0.059149 |
| p-value | - | - | 0.0094 |

**Table 27. ARDL Bounds Testing for Korean data under Fisherian framework.**

| Country | Korea | | |
|---|---|---|---|
| Date Range | 1980-2013 | | |
| Dependent Variable | Lending rate(t) | | |
| Independent Variable | Inflation(t+1) | | |
| Dependent Variable: Max Lag | 5 | | |
| Regressor: Max Lag | 5 | | |
| Fixed Regressors | Constant | Linear Trend | None |
| Selected Model | (1, 1) | (1, 1) | (1, 1) |
| F-Stat (Bound Test) | 9.011595 | 10.18812 | 4.76759 |
| I0 Bound (@5%) | 4.94 | 6.56 | 3.15 |
| I1 Bound (@ 5%) | 5.73 | 7.3 | 4.11 |
| Remark | Long run relationship exists | Long run relationship exists | Long run relationship exists |
| Speed of adjustment | - | - | -0.17959 |
| p-value | - | - | 0.0082 |

**Table 28. ARDL Bounds Testing for Swiss data under Fisherian framework.**

| Country | Switzerland | | |
|---|---|---|---|
| Date Range | 1981-2013 | | |
| Dependent Variable | Lending rate(t) | | |
| Independent Variable | Inflation(t+1) | | |
| Dependent Variable: Max Lag | 5 | | |
| Regressor: Max Lag | 5 | | |
| Fixed Regressors | Constant | Linear Trend | None |
| Selected Model | (2, 5) | (1, 5) | (1, 5) |
| F-Stat (Bound Test) | 2.23778 | 3.943364 | 4.456717 |
| I0 Bound (@5%) | 4.94 | 6.56 | 3.15 |
| I1 Bound (@ 5%) | 5.73 | 7.3 | 4.11 |
| Remark | No long run relationship | No long run relationship | Long run relationship exists |
| Speed of adjustment | - | - | -0.076845 |
| p-value | - | - | 0.0143 |

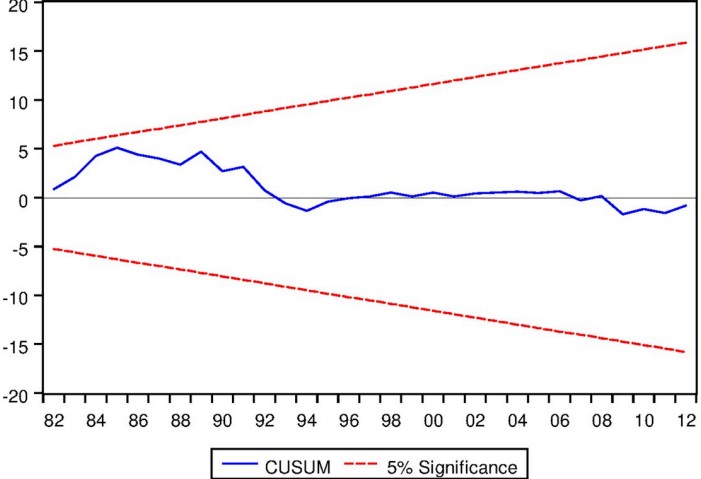

**Fig 25. Stability diagnostic of cointegrating relationship for Australian data with no fixed regressor under Fisherian framework.**

4. Test for the presence of any serial correlation in the aforesaid VAR model. If there is serial correlation amongst the residuals, then increase the lag length until the serial correlation is removed. Let, this lag length be denoted by $p$.

5. Test the dynamic stability of the VAR model having $p$ lags by plotting the inverse roots of the AR characteristic polynomial. The model is said to be stable dynamically, if all the roots lie within the unit circle.

6. Now rebuild the VAR model by adding extra $m$ lags of each of the variables. These additional $m$ lags appear as exogenous in the VAR representation.

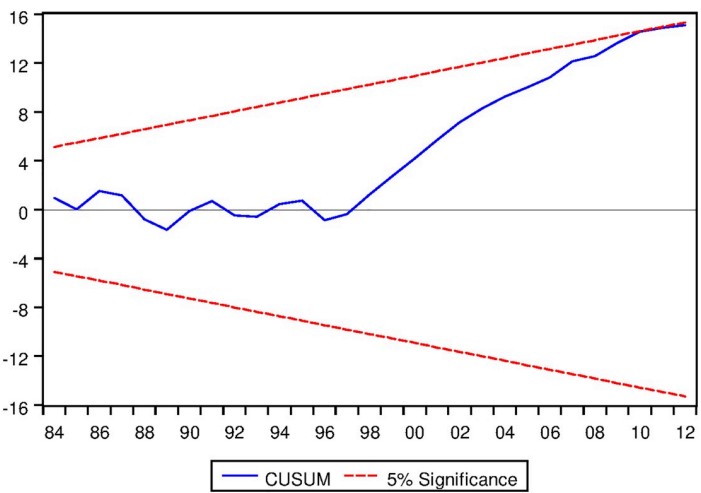

**Fig 26. Stability diagnostic of cointegrating relationship for Japanese data with no fixed regressor under Fisherian framework.**

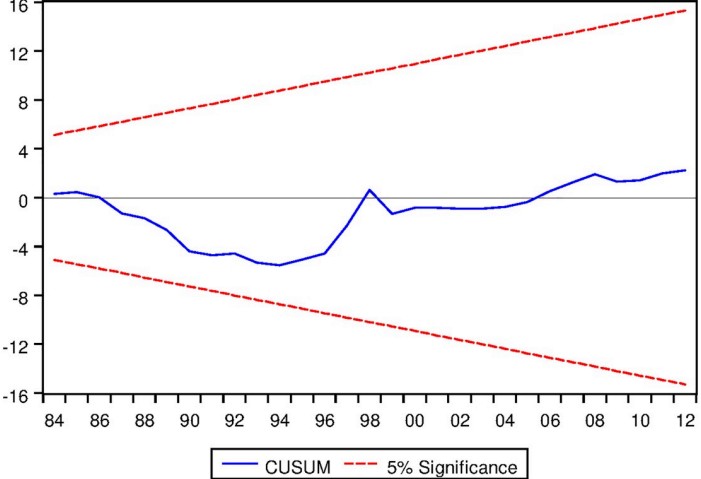

**Fig 27. Stability diagnostic of cointegrating relationship for Korean data with no fixed regressor under Fisherian framework.**

7. On the above manuever of adding $m$ additional lags of each variable in the VAR model as exogenous, the Wald Test Statistics will be asymptotically Chi-square distributed under the null hypothesis of no Granger Causality.

8. Now perform VAR Granger Causality/Block Exogeneity Wald Test and note down the corresponding p-value.

9. The rejection of null hypothesis denotes the existence of Granger Causality amongst the variables.

## 11.2 ADF Unit Root Test

See Tables 1–5 below.

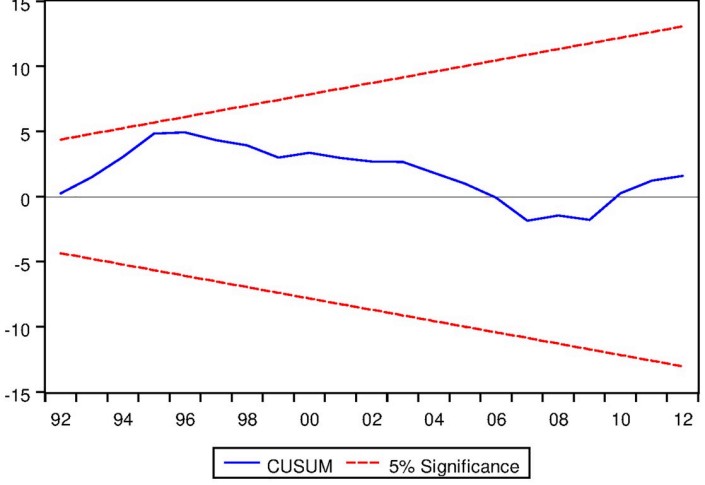

**Fig 28. Stability diagnostic of cointegrating relationship for Swiss data with no fixed regressor under Fisherian framework.**

### 11.3 Lag length selection (proposed model)

See Table 6 below.

### 11.4 Stability diagnostics of the selected VAR model under proposed framework

See Tables 7–11 and Figs 5–9 below.

### 11.5 Lag length selection (Fisherian model)

See Table 12 below.

### 11.6 Stability diagnostics of the selected VAR model under Fisherian framework

See Tables 13–17 and Figs 10–14 below.

### 11.7 VAR Granger Causality/Block Exogeneity Wald Test (proposed model)

See Table 18 below.

### 11.8 VAR Granger Causality/Block Exogeneity Wald Test (Fisherian model)

See Tables 19 and 20 below.

### 11.9 ARDL Bounds Testing under proposed framework

See Tables 21–24 below.

### 11.10 ARDL Bounds Testing under Fisherian framework

See Tables 25–28 below.

### 11.11 Stability diagnostic of cointegrating relationship

See Figs 15–28 below.

## Supporting information

**S1 File. Compiled data.**
(XLSX)

## Acknowledgments

No funding is received to accomplish this work.

## Author Contributions

**Conceptualization:** Ahmed Mehedi Nizam.

**Data curation:** Ahmed Mehedi Nizam.

**Formal analysis:** Ahmed Mehedi Nizam.

**Investigation:** Ahmed Mehedi Nizam.

**Methodology:** Ahmed Mehedi Nizam.

**Writing – original draft:** Ahmed Mehedi Nizam.

**Writing – review & editing:** Ahmed Mehedi Nizam.

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
