## [Decision Letter · Decision Letter 0]

30 Jan 2020

PONE-D-19-33792

How the Banking System is Creating a Two-way Inflation in an Economy?

PLOS ONE

Dear Mr. Nizam,

Thank you for submitting your manuscript to PLOS ONE. After careful consideration, we feel that it has merit but does not fully meet PLOS ONE’s publication criteria as it currently stands. Therefore, we invite you to submit a revised version of the manuscript that addresses the points raised during the review process.

We would appreciate receiving your revised manuscript by Mar 15 2020 11:59PM. To enhance the reproducibility of your results, we recommend that if applicable you deposit your laboratory protocols in protocols.io, where a protocol can be assigned its own identifier (DOI) such that it can be cited independently in the future. For instructions see: http://journals.plos.org/plosone/s/submission-guidelines#loc-laboratory-protocols

We look forward to receiving your revised manuscript.

Kind regards,

Juan Carlos Cuestas

Academic Editor

PLOS ONE

Journal Requirements:

Reviewers' comments:

Reviewer's Responses to Questions

**Comments to the Author**

1. Is the manuscript technically sound, and do the data support the conclusions?

Reviewer #1: Partly

Reviewer #2: Partly

2. Has the statistical analysis been performed appropriately and rigorously? 

Reviewer #1: Yes

Reviewer #2: Yes

3. Have the authors made all data underlying the findings in their manuscript fully available?

Reviewer #1: Yes

Reviewer #2: Yes

4. Is the manuscript presented in an intelligible fashion and written in standard English?

Reviewer #1: Yes

Reviewer #2: Yes

5. Review Comments to the Author

Reviewer #1: General comments

I have found this paper interesting. In my opinion it is well written and the econometric analysis is properly executed to the best of my knowledge. I have a few comments before I can recommend this paper for publication. The main issue is explained in comments 5., 5.1 and 5.2. I am not able to fully assess the relevance of the proposed model because I feel some issues are not well explained.

Specific comments:

1. The Introduction is organized in one single paragraph of 500 words. The introduction is very important in a paper, you should find the way to give a better structure.

2. In section 2, you also have only one paragraph of over 1000 words. It makes more difficult to read your arguments since the reader does not have a natural point to take a break while reading. I think you need to improve readability of section 2 by having shorter paragraphs. Notice this is also in odds with the rest of the paper: sections 3 and 4 have some paragraphs constituted with a single sentence.

3. The review of the literature in section 2 seems also outdated. You do not include any academic paper published since 2002. This does not provide an appropriate framework for your study since it suggests your paper is irrelevant to the current discussion in academia.

4. Is it realistic to set parameter beta as 0<beta<1? 100="" a="" account="" account.="" almost="" also="" any="" bank="" be="" believe="" borrower.="" case="" deposit="" economies="" hard="" in="" is="" it="" loan="" loans="" most="" of="" person="" population="" proportion="" provide="" the="" to="" transferred="" usually="" with="" without="" working="" would="">

5. In pages 8 and 9, you mention the supply and demand functions. You do not clarify what demand/supply function you are referring to. What is Q? This is important because you then have a supply and demand effects which in turn decide inflation. For a while, I thought they were the supply/demand function in the money market (since the Fisher effect is usually modelled from the money demand function). but you also have the APC which suggest an aggregate supply/demand framework.

5.1. There is no explanation why EM affects the demand curve while the EC affects the supply curve. This may be obvious to you but it is not to me. You only mention the effect on your supply-demand function in the last paragraph of page 8. This is the most important comment I have: it is not clear what Q is and why/how EM affects its demand curve while the EC affects its supply curve.

5.2. Related to this, could you explain what would happen if the supply function where to be vertical? this seems relevant for both the money market and AS-AD framework. This question may not be relevant if the supply and demand functions in your paper are something else.

6. Section 6.2 looks like a recipe without much interpretation of the method. I would recommend to move this to an appendix and make use of section 6.2 for you to explain the procedure with your own words. Since the section would most likely be smaller in size you may also consider to eliminate all subsections and move all three methods in section 6 directly.

7. I was surprised in section 7 to find out you have not included the US. This is the most important economy of the world and it is included in most studies of the fisher equation. If the data is available, the US must be part of your study.

8. Can you please clarify what you mean by VAR representations with inflation, X1, X2 as endogenous variables in page 13? You make this reference to VAR with “endogenous variables” many times. To me, this would mean a VAR with three dependent variables and one equation each, which all variables been transformed to I(0). I expect only lags of the dependent variables are included as regressors. This just described a standard VAR. I would like to check with you if I am missing anything from your VAR specifications.

Reviewer #2: The author proposes a new model to study the link between nominal interest rates and inflation with the idea of overcoming the mixed results found when testing for Fisher Effect. The model assumes that nominal deposit and lending interest rates may have different effects on inflation and, also, the intensity of these effects will depend on the volume of deposits and credit.

I have some caveats regarding the paper:

1. The paper assumes that borrowers can translate any increase in interest rates to good prices. However, is this a realistic assumption? There is no reference to the good market structure and the feasibility of this pass-through.

2. The paper analyses a set of very open economies. Is there no role for the exchange rate in the interest rate transmission channel? Are interest rates domestically or international determined? Can lenders and borrowers operate in the international financial markets? If this is the case, the link between interest rate, domestic GDP and domestic inflation may be different to the one described in the paper.

3. Is there any role for structural breaks in the cointegration relationship linking interest rates and inflation (plus some other variables)? Are the cointegration relationships stable? Is it possible to estimate any measure of the degree of mean reversion?

 </beta<1

6. PLOS authors have the option to publish the peer review history of their article (what does this mean?). If published, this will include your full peer review and any attached files.

Reviewer #1: No

Reviewer #2: No

---

## [Author Response · Author response to Decision Letter 0]

14 Feb 2020

Response to Reviewer

Reviewer 1:

1. Introduction is now segregated into three paragraphs: First one sketches the interaction between nominal deposit rate and demand pull inflation. Second paragraph briefly describes the inter-relation between nominal lending rate and cost push inflation. Finally, paragraph 3 combines the arguments presented in paragraph 1 & 2 and discusses how the article is organized. 

2. Section: 2 is now thematically organized around 5 paragraphs each focusing on different clusters of empirical studies regarding Fisher effect.

3. Two more paragraphs about latest state of the art research regarding Fisher effect are appended. Studies earlier than 2002 are cited only to follow the chronological ordering of literature since the inception of Fisher effect in as early as 1930s.

4. Beta represents the portion of credit of those borrowers who do not have deposits with any bank. In practical cases, any customer who has a credit line is also supposed to have a depository account with the bank. Every borrower must have to have either a current/savings/SND account with his/her lending partner. At this point, Beta can be reasonably set to zero. However, the current/savings/SND account of the borrower may have zero balance theoretically. In this hypothetical but plausible scenario, Beta can not be zero. 

5. We are concerned here with the Aggregate Demand – Aggregate Supply (AD-AS) model. Hence, Q represents total output.

5.1. EM and EC are indeed the two sides of the same coin: EM faces the depositors while EC faces the borrowers (producers). EM is actually a portion of nominal interest income while EC corresponds to a portion of nominal interest expense. Like every other transaction, payment of interest involves two parties: One pays and another receives. Hence, changing EM will change the depositors’ income and thus affects aggregate demand curve accordingly. On the other hand, changes in EC will be translated into a change in cost of production (as interest expense is usually considered as a cost of production. See for example, [1], [2], [3]). As cost of production changes so does the aggregate supply.

5.2. Long Run Aggregate Supply (LRAS) is usually considered to be vertical. Hence, although changes in EM and EC may distort the equilibrium in the short run, the economy will eventually lean back to its original full employment level with a different price tag. We have appended a new section (Section: 6) in the article regarding this.

6. We move the step by step algorithm followed for Toda-Yamamoto approach of testing Granger Causality in the context of non-stationary time series to the appendix. We also reorganize ‘Methodology’ section with no sub-section.

7. We omit US from our analysis only because US deposit rate data are not publicly available. We collect annual deposit rate data of commercial banks from World Bank data warehouse and US data are missing there. Moreover, US deposit rate data are not available in other popular public databases like Fed St. Louis economic data, IMF database or OECD statistics. 

8. Reviewer’s understanding in this regard is similar to authors’ one.

Reviewer 2: 

1. Reviewer has rightly pointed out that the borrowers are not the price setters unless it’s a monopoly. We apologize for the ambiguity we created here in the narratives. Our intended meaning in this regard is as follows: When the economy is growing at a rate g, we can assume that the producers of goods and services as a whole also get a g percentage point growth in their production, revenue and profit. If loans in the borrowers' account and output in real terms grow at the same pace (i.e., if g = l) then the accruals in loans can be served effectively by the enhancement in profit. If however g < l then the borrowers have to manage extra money for interest servicing which can not be obtained from the growth in profit. As this is an economy-wide phenomenon not just for one single producer, the aggregate supply curve moves upward accordingly (as we recall interest expense as an inevitable cost of production [1], [2], [3]) and a higher equilibrium price level is set. Assuming a perfect competition, the borrowers/producers just have to follow the new equilibrium price thus set. In this regard, we have completely rewritten Section: 4 of the article.

2. Interest rate, inflation and exchange rate dynamics may be discussed under the following broad heads:

A. Many factors add to interest rate.

B. Interest rate adds to inflation.

C. Inflation adds to exchange rate through international Fisher effect. 

Here, we confine ourselves to point B. Detailing point A and C simultaneously along with point B may go beyond the scope of a single study. 

3. a) In the current study, structural breaks in the time series data are not taken into account. Incorporating structural break in our analysis at this point will irrevocably nullify all the testings that have been done, from unit roots to Block Exogeneity to ARDL. The study will then require new estimation techniques and may pose different results and interpretations which we think will go beyond the scope of a “Revise and Resubmit”.

b) Now, we have reported the results of CUSUM test to judge the stability of the selected ARDL models which were not included in the previous submission. Results suggest that most of the ARDL models are stable.

c) Now, we have reported the speed of adjustment and corresponding p-values for our selected ARDL models. Speed of adjustments are found to be negative in all the cases which implies some sort of mean reversion in the process and the corresponding p-values are found to be significant in almost all the reported cases of cointegration. 

References: 

[1] Barth, M. and V. Ramey, 2001, The Cost Channel of Monetary Transmission, NBER

Macroeconomics Annual 2001, Cambridge, MIT Press, pp. 199-240.

[2] Dedola, L., Lippi, F., 2005, The monetary transmission mechanism: evidence from

the industries of five OECD countries, European Economic Review 49, 1543-1569.

[3] Gaiotti, E., Secchi, A. (2006), Is there a cost channel of monetary policy transmission? An investigation into the pricing behaviour of 2,000 firms, Journal of Money, Credit and Banking, v. 38, 8, pp. 2013-2038.

---

## [Editor Report · Decision Letter 1]

19 Feb 2020

How the Banking System is Creating a Two-way Inflation in an Economy?

PONE-D-19-33792R1

Dear Dr. Nizam,

We are pleased to inform you that your manuscript has been judged scientifically suitable for publication and will be formally accepted for publication once it complies with all outstanding technical requirements.

With kind regards,

Juan Carlos Cuestas

Academic Editor

PLOS ONE
---

## [Editor Report · Acceptance letter]

24 Feb 2020

PONE-D-19-33792R1 

How the Banking System is Creating a Two-way Inflation in an Economy? 

Dear Dr. Nizam:

I am pleased to inform you that your manuscript has been deemed suitable for publication in PLOS ONE. Congratulations! Your manuscript is now with our production department. 

With kind regards,

on behalf of

Professor Juan Carlos Cuestas 

Academic Editor

PLOS ONE